# Towards Multi-Domain Learning for Generalizable Video Anomaly Detection

**MyeongAh Cho**[*]
Kyung Hee University
maycho@khu.ac.kr

**Taeoh Kim**
NAVER Cloud
taeoh.kim@navercorp.com

**Minho Shim**
NAVER Cloud
minho.shim@navercorp.com

**Dongyoon Wee**
NAVER Cloud
dongyoon.wee@navercorp.com

**Sangyoun Lee**[†]
Yonsei University
syleee@yonsei.ac.kr

## Abstract

Most of the existing Video Anomaly Detection (VAD) studies have been conducted within single-domain learning, where training and evaluation are performed on a single dataset. However, the criteria for abnormal events differ across VAD datasets, making it problematic to apply a single-domain model to other domains. In this paper, we propose a new task called **M**ulti-**D**omain learning for **VAD** (**MDVAD**) to explore various real-world abnormal events using multiple datasets for a general model. MDVAD involves training on datasets from multiple domains simultaneously, and we experimentally observe that *Abnormal Conflicts* between domains hinder learning and generalization. The task aims to address two key objectives: *(i)* better distinguishing between *general* normal and abnormal events across multiple domains, and *(ii)* being aware of ambiguous abnormal conflicts. This paper is the first to tackle abnormal conflict issue and introduces a new benchmark, baselines, and evaluation protocols for MDVAD. As baselines, we propose a framework with Null(Angular)-Multiple Instance Learning and an Abnormal Conflict classifier. Through experiments on a MDVAD benchmark composed of six VAD datasets and using four different evaluation protocols, we reveal abnormal conflicts and demonstrate that the proposed baseline effectively handles these conflicts, showing robustness and adaptability across multiple domains.

## 1 Introduction

Video Anomaly Detection (VAD) is identifying abnormal events in diverse scenarios depicted in a video and determining their temporal intervals at a frame-level. Nowadays, CCTVs are ubiquitous, recording every moment of life, which aids in preventing accidents and responding to crimes promptly. However, human monitoring of every situation is highly inefficient, requiring significant labor and resources. Consequently, extensive research has been conducted to automate VAD through deep learning by leveraging large amounts of surveillance data [33, 5, 24, 15, 55, 44, 43, 50, 21, 7].

In VAD research, Weakly-supervised VAD (WVAD) [55, 44, 43, 50, 21, 7], which involves learning normal and abnormal events with minimal supervision of the video-level annotation and detecting abnormal events at the frame-level during testing, has been studied a lot recently. This paper focuses on the WVAD setting (denoted as VAD), and a summary of VAD research is explained in the

---

[*]Work done while doing an internship at NAVER Cloud.

[†]Corresponding author

38th Conference on Neural Information Processing Systems (NeurIPS 2024).

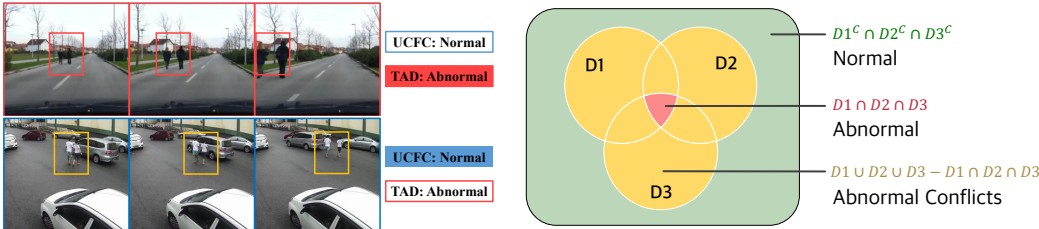

(a) Examples of Abnormal Conflict between datasets      (b) Venn diagrams of events

Figure 1: **(a)** An example of abnormal conflict: *Pedestrian on the road* is normal in UCFC dataset but is abnormal in TAD. **(b)** Each circle represents each domain. MDVAD aims to design a general model that effectively considers abnormal conflicts to separate general normal and abnormal events.

supplementary material (§ E). Unlike the conventional VAD research, we first address the following three key questions:

***Q1: What is the problem with the existing VAD model?*** Most VAD models are trained in a single-domain where the training and evaluation dataset are the same. In the case of single-domain learning, application across different datasets (cross-domain evaluation) results in performance degradation as reported in [14, 6, 7] because VAD models are heavily influenced by the criteria for abnormality defined by each dataset.

***Q2: Why do we need a general VAD model?*** First, a single generalized model removes the need for multiple specific models for different domains, analogous to multi-task learning. Second, proper pre-training on multiple domains embodies generalized representation, and it can discriminate abnormal events according to their domain, leading to better performance in unseen target domains. Consequently, a general VAD model will be highly beneficial for applying VAD in practical scenarios.

***Q3: Is it possible to create a general VAD model?*** The general VAD model aims to handle multiple domains, but this is challenging because the definition of abnormal differs for each dataset, leading to conflicts between these abnormal events. For example, as shown in Fig. 1(a), in one dataset, a pedestrian on the road is considered normal, while in another dataset, it is deemed abnormal. A general VAD cannot be solved with a naive muti-task learning because of this confusion across multiple domains, which this paper defines as **'Abnormal Conflict.'** Therefore, for general VAD, it is necessary to be aware of these abnormal conflicts (yellow region in Fig. 1(b)) and learn general normal (green region in Fig. 1(b)) or abnormal (red region in Fig. 1(b)) representations that are common across all domains.

Our goal is to construct a general VAD model by conducting **multi-domain learning** while recognizing **abnormal conflicts** and exploring representations of **general normality and abnormality**. To achieve this goal, we introduce a new task called **1) Multiple Domain VAD (MDVAD), along with a benchmark and new evaluation protocols.** MDVAD involves concurrent training on multiple VAD datasets, each with its own definition of abnormality. Specifically, the MDVAD benchmark comprises six representative VAD datasets with balanced sampling (§ 4.1). We also propose four evaluation protocols: held-in, leave-one-out, low-shot domain adaptation, and full fine-tuning. The held-in protocol is designed to evaluate the model's ability as a unified model like multi-task model, while the leave-one-out, low-shot domain adaptation, and full fine-tuning protocols are intended to access the model's capability as a general pre-training for an unseen target domain.

As multi-domain learning is a novel concept in the field of VAD, we also introduce baselines and new learning methods. We design domain-specific multiple heads to mitigate abnormal conflicts and learn generality across domains. To facilitate multi-head learning without conflicts, we propose the **2) Null-Multiple Instance Learning (Null-MIL) and NullAngular-MIL (NullAng-MIL) losses**, which activate only the output of the head corresponding to the input domain, assigning inactive heads with *Null* values to prevent confusion. Additionally, we suggest the **3) Abnormal Conflict (AC) Classifier to explore general features while being aware of abnormal conflicts**, leveraging the variance in abnormal scores across the multiple heads (Fig. 2 green and yellow region in the Venn diagram). Through experiments with four protocols on the MDVAD benchmark, we reveal the limitations of multi-domain learning with abnormal conflict and demonstrate the effectiveness of our baselines in offering a generalized and adaptive model.

## 1.1 Scope of research

**Focusing on Multiple Domains.** In this paper, we focus on solving the problems in multiple domains introduced above rather than solving problems in a single domain. Therefore, complex design of the backbone and head or achieving state-of-the-art performance in a single domain can proceed orthogonally with our research. Instead, this paper raises the issue of abnormal conflict and focuses on the necessity of MDVAD.

**Distinction from Open-set VAD Approaches.** Unlike Open-set VAD [59, 25, 1], which separates seen and unseen anomalies within a single-domain cannot achieve robust representation learning among multiple domains, this paper excels at handling abnormal conflicts and shows adaptability across domains with versatile evaluation protocols (held-in/out and low-shot).

# 2 Observations

## 2.1 Datasets

In this paper, we use six representative VAD datasets: UCF-Crimes (UCFC) [43], XD-Violences (XD) [51], Large-scale Anomaly Detection (LAD) [47], UBI-Fights (UBIF) [9], Traffic Anomaly Dataset (TAD) [19], and Shanghai-Tech Campus (ST) [24]. As shown in Table 1, each dataset has different environments (*e.g.* CCTV, Traffic, Campus), quantities, and abnormal categories. Unlike other datasets, ST is an unsupervised VAD benchmark whose training set comprises only normal videos, so the training set has been reorganized following [21, 45, 57]. More details are provided in the supplementary material (Sec. A).

## 2.2 Analysis

Based on observations of each dataset's properties, we aim to quantify how the different properties negatively affect domain shifts. Table 2 presents cross-domain evaluation results, *i.e.*, single-domain models validated on different target datasets. The results reveal that while these models excel in-domain settings (diagonal elements of the table), they exhibit significant performance degradation in cross-domain scenarios. This implies that a single-domain VAD model may be ineffective in most other environments unless the environment and user's intention are precisely identical. Therefore, leveraging diverse datasets for generalized feature learning is crucial, enabling the model to handle unknown domain and well adapted to unseen anomalies. This paper addresses two primary issues: 1) Abnormal conflict arising during the multi-domain learning process, and 2) Scene discrepancy occurring during the evaluation process on an unseen target domain.

**Abnormal conflict.** As illustrated in Fig. 1, abnormal conflicts indicate abnormal events that are considered abnormal in one (or some) domain(s) but are denoted as normal in other domains. As

Table 1: Datasets and abnormal categories. Colored categories are shared abnormal categories with other datasets. Uncolored categories are considered normal events in other datasets. Gray categories are defined by ourselves while all other categories are provided by datasets.

| UCFC [43] | XD [51] | LAD [47] | UBIF [9] | TAD [19] | ST [24] |
|---|---|---|---|---|---|
| **Settings** | | | | | |
| CCTV | CCTV, Sports, Cartoon, Movies, News | CCTV | CCTV, Mobile | Traffic (CCTV) | Campus (CCTV) |
| **Training Set Volume** | | | | | |
| 1610 | 3954 | 1440 | 933 | 400 | 238 |
| **Abnormal Categories** | | | | | |
| Abuse | Abuse | Hurt | | Illegal Turns | Chasing |
| Arrest | Drop | Fall Into Water | | Road Spills | Dropping |
| Arson | | Falling | | Retrograde Motion | Throwing |
| Assault | | Crash | | Illegal Occupations | Running |
| Accident | Car Accident | Destroy | | Vehicle Accidents | Jumping |
| Burglary | | Fire | | Pedestrian on Road | Motorcycle |
| Explosion | Explosion | | | The Else | Skateboard |
| Fighting | Fighting | Fighting | Fighting | | Fighting |
| Robbery | | Violence | | | Robbery |
| Shooting | Shooting | Crowd | | | Gun |
| Stealing | | Thieving | | | Car |
| Shoplifting | | Loitering | | | Loitering |
| Vandalism | Riot | Panic | | | |
| | | Trampled | | | |

Table 2: Anomaly detection performances (Area under curve, AUC) of single-domain models. Diagonal elements are in-domain results and off-diagonal elements are cross-domain results.

| | Target | | | | | |
|---|---|---|---|---|---|---|
| Source | UCFC | XD | LAD | UBIF | TAD | ST |
| UCFC | 82.32 | 68.06 | 75.75 | 71.12 | 73.75 | 59.24 |
| XD | 68.38 | 90.87 | 77.60 | 67.23 | 71.10 | 46.87 |
| LAD | 59.60 | 75.26 | 86.97 | 59.27 | 73.80 | 47.29 |
| UBIF | 74.79 | 75.22 | 70.29 | 93.63 | 68.16 | 54.21 |
| TAD | 50.83 | 45.38 | 52.02 | 61.95 | 90.71 | 41.58 |
| ST | 55.75 | 52.87 | 48.96 | 59.00 | 48.57 | 91.88 |

Table 3: Scene discrepancy between domains (Earth mover's distance, EMD), Yellow: Abnormal Class-wise. Blue: Normal. Using a pretrained backbone model on the Kinetics dataset [4], we extract embedding features for each dataset and measured the distance between feature vectors.

| | UCFC | XD | LAD | UBIF | TAD | ST |
|---|---|---|---|---|---|---|
| UCFC | - | 7.90 | 8.45 | 5.87 | 9.76 | 9.32 |
| XD | 9.76 | - | 6.19 | 6.94 | 9.65 | 9.10 |
| LAD | 12.78 | 10.76 | - | 7.90 | 10.09 | 10.43 |
| UBIF | 7.90 | 7.37 | 11.99 | - | 10.54 | 9.54 |
| TAD | 11.32 | 10.87 | 14.39 | 10.98 | - | 10.65 |
| ST | 11.32 | 10.31 | 14.27 | 8.77 | 11.77 | - |

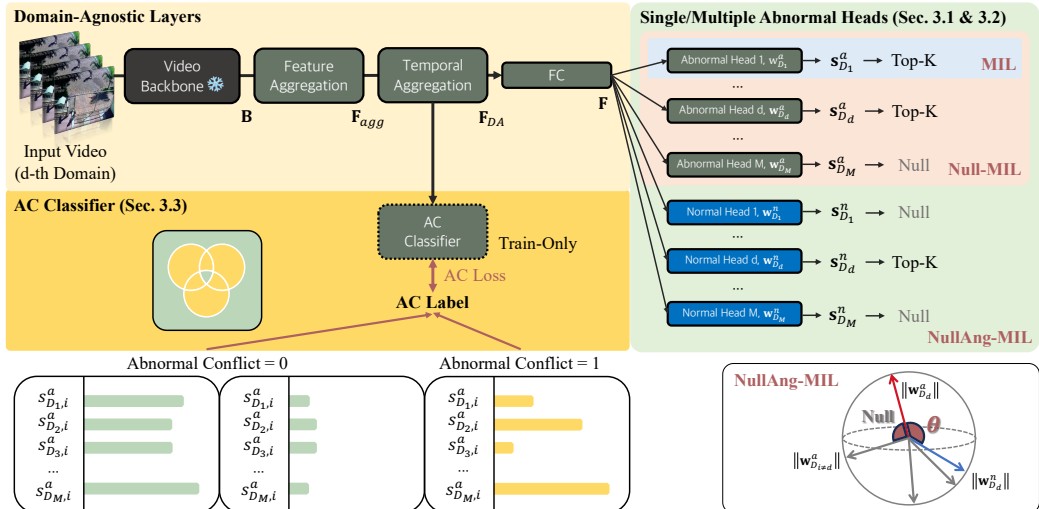

Figure 2: The overall framework of our MDVAD baselines that consists of domain-agnostic layers, single abnormal head (Sec. 3.1), multiple abnormal heads (Sec. 3.2), and AC classifier (Sec. 3.3).

shown in Table 1, there is relatively little overlap of abnormal classes in ST with other datasets. In other words, this means that the abnormal conflict with other datasets is relatively large. This conflict leads to low cross-dataset performances of models trained or evaluated on ST in Table 2. Taking UCFC as an example (target domain: UCFC column in Table 2), performance increases in the order of TAD, ST, LAD, XD, and UBIF, which is proportional to the number of abnormal conflict categories that exist in the source domain. Abnormal conflict makes the MDVAD unique as the label spaces between domains literally conflict because of each other's differing definitions.

**Scene discrepancy.** Scene discrepancy refers to differences in the visual settings of scenes, distinct from abnormal conflict arising from variations in the definition of abnormal classes. To quantify scene discrepancy, we utilize the Earth Mover's Distance (EMD) [38] introduced in [8] to calculate the distance between VAD datasets in the Table 3. In Table 3, the top-right section illustrates the comparison of normal features, while the bottom-left provides a numerical comparison of class-wise abnormal features. Lighter colors indicate a higher discrepancy between datasets. Unlike other datasets, TAD, which comprises traffic videos, exhibits a large distance from normal sample distances in the dataset, while LAD, with diverse and complex scenes for abnormal classes, shows the furthest distance from other datasets. This explains the results of domain adaptation experiments. (Table 7).

## 3 Baselines

Fig. 2 depicts the overall framework of our MDVAD baselines divided into domain-agnostic layers, consisting of the video backbone and aggregation modules, Single (§ 3.1) or Multiple heads (§ 3.2), and AC classifier (§ 3.3).

**Domain-agnostic layers.** The input abnormal or normal video $\mathbf{V}^a$ or $\mathbf{V}^n$ is divided into uniformly sampled $T$ snippets ($\mathbf{V}^a \in \{\mathbf{v}_1^a, \cdots, \mathbf{v}_T^a\}$). These $T$ snippets pass through a pre-trained video backbone, resulting in a $C$-dimensional feature $\mathbf{B} \in \mathbb{R}^{T \times C}$ that undergoes an aggregation module, fusing them from the feature level to the temporal level. The feature aggregation layer doubles the channel size $\hat{\mathbf{B}} \in \mathbb{R}^{T \times 2C}$, followed by a split and max operation $\mathbf{F}_{agg,i} = \max\left[\hat{\mathbf{B}}_i^c, \hat{\mathbf{B}}_i^{C+c}\right]_{c=1,\cdots,C}$ to squeeze the channel size back to $C$ for the $i$-th snippet. Activating only the maximum element during gradient propagation enables the implicit differentiation of class-specific channels, allowing the model to establish discrepancies between classes [7, 52]. The temporal aggregation layer, with a temporal kernel size of 3, produces the domain-agnostic aggregated feature $\mathbf{F}_{DA} \in \mathbb{R}^{T \times C/2}$.

## 3.1 Single-domain learning

The final feature $\mathbf{F}_{DA}$ undergoes fully connected layers denoted as FC and the single abnormal head ($\mathbf{w}_{D_1}^a \in \mathbb{R}^{C/16 \times 1}$) followed by a sigmoid function to derive the final abnormal score $\mathbf{s}_{D_1}^a \in \mathbb{R}^{T \times 1}$.

**MIL.** Due to the absence of temporal interval training labels for abnormal events, WVAD models rely on video-level labels for training, employing the Multiple Instance Learning (MIL) method. When the top-$K$ score set is represented as $\Omega_k(\mathbf{s}_{D_1}^a)$, the Binary Cross Entropy-based (BCE) classification loss function is formulated as presented in Eq. 1, where $y = \{0, 1\}$.

$$L_{MIL} = \sum_{i \in \Omega_k(\mathbf{s}_{D_1}^a)} -(y \log s_{D_1,i}^a + (1-y) \log(1 - s_{D_1,i}^a)), \tag{1}$$

This loss function ensures that only snippets with high (Top-$K$) abnormal scores can contribute to the loss. This single-head model serves as the single-domain baseline (MIL in Fig. 2).

## 3.2 Multi-domain learning: Multi-head learning

**Null-MIL.** To address the abnormal conflict issue mentioned in Sec. 2, in the MDVAD framework, the abnormal head is divided into multiple heads, each responsible for its own domain, allowing for the domain-wise prediction of the output score. Inspired by [18], the prediction score for the input snippet from the $D_d$ dataset is derived exclusively from the output of the $D_d$-head, and the results from heads of other datasets are filled with *Null* values (Null-MIL in Fig. 2). Compared to Eq. 1, the Null-MIL loss is changed as follows:

$$L_{Null-MIL} = \sum_{d=1}^{M} \sum_{i \in \Omega_k(\mathbf{s}_{D_d}^a)} -(y \log s_{D_1,i}^a + (1-y) \log(1 - s_{D_1,i}^a)) \tag{2}$$

where $M$ is the number of heads (domains). To avoid the abnormal conflicts, the heads between datasets are separated where $D_d$-head's weight $\mathbf{w}_{D_d}^a$ is independently trained for the corresponding dataset. In Eq. 2, only $\mathbf{s}_{D_d}^a$ among all output scores is added to the loss, thus the gradient becomes $\frac{\partial \mathbf{s}_{D_d}^a}{\partial \mathbf{w}_{D_d}^a}$, while other heads are not affected.

For the test, the abnormal score of $i$-th snippet is $s_{D_d,i}^a$ when the target dataset is $D_d$ and $\max_d s_{D_d,i}^a$ by selecting the maximum value for unseen target data.

**NullAng-MIL.** We additionally propose a NullAngular-based MIL method that employs the angular margin to effectively diminish large variations among intra-class instances. In this case, multiple normal heads are added (NullAng-MIL in Fig. 2). When head classifier weight of each dataset $D_d$ is denoted as $\mathbf{w}_{D_d}^a$ and $\mathbf{w}_{D_d}^n$ and final embedding feature is $\mathbf{F}$, the final abnormal and normal scores are represented by $\mathbf{s}_{D_d}^a = \mathbf{F} \cdot \mathbf{w}_{D_d}^a$ and $\mathbf{s}_{D_d}^n = \mathbf{F} \cdot \mathbf{w}_{D_d}^n$, respectively. Normalizing the head weight and feature vector to 1 results in $\mathbf{s}_{D_d}^a = \|\mathbf{F}\| \|\mathbf{w}_{D_d}^a\| \cos \boldsymbol{\theta}_{D_d}^a = \cos \boldsymbol{\theta}_{D_d}^a$ and $\mathbf{s}_{D_d}^n = \cos \boldsymbol{\theta}_{D_d}^n$, representing cosine similarity. Thus, in the cosine space, Eq. 2 can be defined as Eq. 3, requiring the maximum cosine similarity between the feature from dataset $D_d$ and the abnormal head $\mathbf{w}_{D_d}^a$ to be at least an angular margin of $m$ greater than the normal head $\mathbf{w}_{D_d}^n$ that enlarging the gap between normal and abnormal.

$$\max_i \cos(\theta_{D_d,i}^a + m) > \max_i \cos \theta_{D_d,i}^n \tag{3}$$

Denoting the top-$K$ abnormal scores from the head of dataset $D_d$ as $\Omega_k(\mathbf{s}_{D_d}) = \{\mathbf{s}_{D_d,i}^a, \mathbf{s}_{D_d,i}^n\}_{i=\text{topk indices}}$, rewriting Eq. 2 as an angular margin-based regression problem results in Eq. 4. Similar to Null-MIL, the loss is computed by the head associated with the input dataset, while scores from other heads have no impact on updating the model's weights.

$$L_{NullAng-MIL} =$$
$$\sum_{d=1}^{M} \sum_{i \in \Omega_k(\mathbf{s}_{D_d})} -\left(y \log \frac{e^{\cos(\theta_{D_d,i}^a + m)}}{e^{\cos(\theta_{D_d,i}^a + m)} + e^{\cos \theta_{D_d,i}^n}}\right.$$
$$\left. + (1-y) \log \frac{e^{\cos(\theta_{D_d,i}^n + m)}}{e^{\cos \theta_{D_d,i}^a} + e^{\cos(\theta_{D_d,i}^n + m)}}\right) \tag{4}$$

For the test, since the normal and abnormal heads are trained through angular margin learning, the abnormal score is shown in Eq. 5.

$$\text{Abnormal Score}_i$$
$$= \begin{cases} s^a_{D_d,i} + (1 - s^n_{D_d,i}) & \text{source } D_d = \text{target} \\ \max_d s^a_{D_d,i} + (1 - \max_d s^n_{D_d,i}) & \text{source} \neq \text{target} \end{cases} \tag{5}$$

When the source domain of the pre-trained general model is different from the target domain, we determine the final score to reflect conflicts by taking the maximum normal and abnormal score from multiple heads.

**Unseen domain adaptation.** After multi-domain learning, when a new target dataset or unseen condition appears, the final score is computed based on the similarity between the embedding feature of the input video and each source domain's head classifier. Therefore, it operates by considering domain similarity, which can address the performance degradation issue that occurs due to the scene discrepancy discussed in Sec. 2.

**Complexity.** Only the final layer is added based on the number of datasets, with the head's weight denoted as $\mathbf{w}_{D_d} \in \mathbb{R}^{T \times 1}$, which is a very small proportion of the entire model. Comparing a single and multiple heads (6 datasets), the training times are 2.68 and 2.81 hours, and the inference times are 0.158 and 0.164 milliseconds per snippet, respectively, indicating a negligible increase in complexity.

### 3.3 Abnormal Conflict (AC) classifier

In the domain-agnostic feature extraction phase, it is crucial to capture general features that can handle every domain. While we have divided the heads to avoid abnormal conflicts in multiple source datasets, the agnostic part extracts features from all datasets using a single branch, and inconsistent labels cause confusion. Therefore, we propose an Abnormal Conflict (AC) classifier for learning the final embedding feature $\mathbf{F}_{DA}$ that passes through the detector heads.

In Fig. 2, each classifier head distinguishes between $D_d$ and $D_d^c$, while the AC classifier is intended to distinguish between elements that are abnormal or normal across all source datasets (green area) and elements that represent conflicts (yellow area) within the Venn diagram. The AC classifier takes the embedding feature $\mathbf{F}_{DA}$ as input, followed by two FC layers, to predict the final conflict score $\mathbf{s}^{AC}$. In Eq. 6, the AC label is generated based on the scores of all abnormal heads, where $y_i^{AC} = 1$ if the deviation between the scores is above a threshold $\tau$.

$$y_i^{AC} = \begin{cases} 1 & [\max_d s^a_{D_d,i} - \min_d s^a_{D_d,i} - \tau]_+ > 0 \\ 0 & \text{otherwise} \end{cases} \tag{6}$$

The loss of the AC classifier, denoted as $L_{AC}$ and calculated using cross-entropy as follows:

$$L_{AC} = \sum_{i=1}^{T} -(y_i^{AC} \log s_i^{AC} + (1 - y_i^{AC}) \log(1 - s_i^{AC})) \tag{7}$$

The total objective function is $L = L_{NullAng-MIL} + \lambda L_{AC}$.

During the testing phase, the auxiliary branch AC classifier is eliminated, and the output is calculated for each input snippet $\mathbf{v}_i$ as the final Abnormal Score$_i$.

## 4 Experimental Results

### 4.1 MDVAD benchmark

As shown in Table 1, VAD comprises six representative datasets with diverse settings and volumes. The MDVAD is a task aimed at addressing domain shifts between datasets, and each dataset included in the MDVAD benchmark should be structured so that it is not biased toward any particular dataset or anomalies. Consequently, datasets should have an equal volume in the training set and encompass various abnormal categories and criteria. To achieve this, we sampled each dataset to align with the dataset with the smallest volume, ensuring that each abnormal category has a similar proportion

Table 4: Single-domain results (AUC): In-domain (diagonal elements) and cross-domain (off-diagonal elements) results.

| | MDVAD Benchmarks | | | | | |
|---|---|---|---|---|---|---|
| Source | Target | | | | | |
| | UCFC | XD | LAD | UBIF | TAD | ST |
| UCFC | 77.93 | 67.07 | 74.23 | 71.10 | 67.95 | 50.84 |
| XD | 65.88 | 83.23 | 72.63 | 70.58 | 63.46 | 49.55 |
| LAD | 60.23 | 71.37 | 83.82 | 64.73 | 68.47 | 47.27 |
| UBIF | 74.11 | 70.49 | 67.69 | 92.62 | 68.03 | 56.47 |
| TAD | 56.22 | 45.05 | 58.67 | 67.83 | 90.75 | 41.92 |
| ST | 50.49 | 61.53 | 60.91 | 56.58 | 41.60 | 90.79 |
| Out Avg. | 61.39 | 63.10 | 66.83 | 66.17 | 61.90 | 49.21 |
| (In-domain) | (77.93) | (83.23) | (83.82) | (92.62) | (90.75) | (90.79) |

**Training:** Single-source dataset / **Testing:** Target dataset
**Out Avg.:** Average of cross-domain results.

Table 5: **E1:** Multi-domain training: held-in results (AUC).

| | | MDVAD Benchmarks | | | | | | |
|---|---|---|---|---|---|---|---|---|
| | | Target | | | | | | Avg. |
| | | UCFC | XD | LAD | UBIF | TAD | ST | |
| *Single-domain* | | | | | | | | |
| Out Avg. | | 61.39 | 63.10 | 66.83 | 66.17 | 61.90 | 49.21 | 61.43 |
| (In-domain) | | (77.93) | (83.23) | (83.82) | (92.62) | (90.75) | (90.79) | (86.52) |
| *E1: Held-in* | | | | | | | | |
| Head | AC | UCFC | XD | LAD | UBIF | TAD | ST | Avg. |
| MIL | − | 80.05 | 83.77 | 86.01 | 85.76 | 88.92 | 88.82 | 85.56 |
| | ✓ | 80.11 | 83.91 | 85.15 | 87.72 | 90.05 | 87.98 | 85.82 |
| Null-MIL (Ours) | − | 79.01 | 81.96 | 85.08 | 93.06 | 90.57 | 91.04 | 86.79 |
| | ✓ | 79.15 | 82.96 | 85.82 | 92.41 | 91.16 | 89.67 | 86.86 |
| NullAng MIL(Ours) | − | 76.32 | 82.74 | 82.32 | 92.30 | 91.82 | 91.26 | 86.13 |
| | ✓ | 77.21 | 82.09 | 83.88 | 91.90 | 91.36 | 91.12 | 86.26 |

**Training:** All six datasets / **Testing:** Target dataset
Column-wise coloring with increased intensity for higher values

and conducted 3-fold experiments. Additionally, to handle small datasets like TAD and ST, which have minimal volumes, we combined them with the traffic dataset CADP [41] and campus dataset NWPU [3], respectively, by reorganizing their training sets. The volume of MDVAD benchmark is unified at 386 videos per dataset. More details provided in the supplementary material (Sec. B).

## 4.2 Empirical studies

This section presents empirical studies as follows: first, single-domain results for comparison, followed by the results of four evaluation protocols for the MDVAD benchmark. The models are trained on the training set of the source domain and evaluated on the test set of the target domain under the following settings: Held-in (E1), Leave-one-out (E2), Low-shot adaptation (E3), and Full fine-tuning (E4). The Area Under Curve (AUC) is used as the evaluation metric. The hyper-parameters are $T = 32$, $\lambda = 10$, $\tau = 0.3$, and $m = 0.3$. Since the single-head baseline cannot assign AC labels using Eq. 6, pseudo-labels for the AC classifier are assigned based on the range of predicted abnormal scores. All detailed implementations, more results, and discussions are provided in the supplementary material.

**Single-domain results.**    In the preliminary experiment, we examine the performance of the MDVAD benchmark under single-source and single-target conditions using a single-head MIL baseline. Comparing the in-domain (diagonal elements) results in Table 2 (using the original training set) and Table 4 (using the sampled training set), we observe that the performance diminishes when datasets, other than TAD and ST, are reduced during sampling. Conversely, TAD and ST show improved performance, benefiting from increased diversity and data volume due to reorganization with CADP and NWPU for TAD and ST, respectively. The cross-domain (off-diagonal elements) results in Table 4 demonstrate consistent trends with both the MDVAD and VAD benchmarks. These trends primarily arise from challenges related to abnormal conflict and scene discrepancies, indicating that the issues are confined to in-domain settings. Consequently, to uphold in-domain performance across diverse datasets and effectively address issues related to abnormal conflict and scene discrepancies, we emphasize the essential role of multi-domain learning in developing a general model for VAD.

### 4.2.1 Held-in results (E1)

In this section, we discuss the held-in evaluation results, where models are trained on all datasets of MDVAD, and the test set of each source dataset is used as the target. Table 5 presents the results obtained by training with the MIL baseline with a single-head, which is commonly used in traditional WVAD, and the proposed Null(Ang)-MIL baseline composed of multiple heads. Because UCFC, XD, and LAD exhibit low abnormal conflicts, numerous similarities in abnormal categories, and minimal scene discrepancy, there are great performances with the single-head MIL baseline. However, when TAD and ST are targets, the results demonstrate that the Null(Ang)-MIL baselines, which extract general domain features in the domain-agnostic part and avoid conflict by assigning Null values with multiple heads, outperform the MIL baseline.

**Handling multi-domain with a single Null(Ang)-MIL baseline.** The average performance of single Null-MIL and NullAng-MIL is comparable to or even better than the average performance of individual in-domain models trained for each domain (86.52%). Moreover, the NullAng-MIL baseline, which effectively performs inter/intra-class learning through cosine angular margin while avoiding conflict between each head, achieves superior results, with 91.82% (TAD) and 91.26% (ST) compared to the performances of single in-domain results of 90.75% (TAD) and 90.79% (ST),

Table 6: **E2**: Leave-one-out results

| MDVAD Benchmarks | | | | | | | |
|---|---|---|---|---|---|---|---|
| | | Target | | | | | |
| | | UCFC | XD | LAD | UBIF | TAD | ST |
| *Single-domain* | | | | | | | |
| Out Avg. | | 61.39 | 63.10 | 66.83 | 66.17 | 61.90 | 49.21 |
| *E2: Leave-one-out* | | | | | | | |
| Head | AC | | | | | | |
| MIL | – | 75.98 | 74.07 | 76.94 | 72.01 | 74.11 | 49.39 |
| | ✓ | 78.49 | 76.87 | **78.67** | 81.81 | 78.39 | 65.66 |
| Null-MIL | – | 62.38 | 59.63 | 64.91 | 55.42 | 66.28 | 45.60 |
| **(Ours)** | ✓ | 68.78 | 74.65 | 74.46 | 55.61 | 67.72 | 55.26 |
| NullAng | – | 75.26 | 73.00 | 73.91 | 79.41 | 77.94 | 52.98 |
| MIL**(Ours)** | ✓ | **78.55** | **77.68** | 77.36 | **82.53** | **79.21** | **60.41** |

**Training:** Five datasets except the target dataset
**Testing:** Target dataset

Table 7: **E3**: Low-shot adaptation results

| MDVAD Benchmarks | | | | | | | |
|---|---|---|---|---|---|---|---|
| | | Target | | | | | |
| | | UCFC | XD | LAD | UBIF | TAD | ST |
| *E3: Low-shot Adaptation* | | | | | | | |
| Head | AC | | | | | | |
| MIL | – | 75.19 | 68.20 | **79.18** | 82.13 | 82.80 | 71.65 |
| | ✓ | 72.52 | 71.00 | 76.69 | 82.34 | 78.72 | 74.88 |
| Null-MIL | – | 67.55 | 60.32 | 75.11 | 75.97 | 62.29 | 57.72 |
| **(Ours)** | ✓ | 70.57 | 66.40 | 73.58 | 81.39 | 71.12 | 63.02 |
| NullAng | – | 77.76 | 70.67 | 74.86 | 83.44 | 78.57 | 71.81 |
| MIL**(Ours)** | ✓ | **78.99** | **75.80** | 77.82 | **85.75** | **84.06** | **76.23** |

**Training:** E2 + a few target samples
**Testing:** Target dataset

highlighting the effectiveness of learning with diverse and abundant data to enhance model generality. Furthermore, additional performance boosts are observed across all baselines with the auxiliary AC classifier to make the model conflict-aware.

#### 4.2.2 Leave-one-out results (E2)

The E2 experiments, in Table 6, are conducted on the leave-one-out setting among multiple source datasets and evaluated on the unseen target dataset unlike E1. Each column represents the outcomes of models trained with the target dataset excluded. Compared with the held-in results, the evaluation results conducted without prior knowledge of abnormal boundaries for unseen target data lead performance gap between E1 and E2.

**Effectiveness of multi-domain learning for unseen target domain.** When comparing single-domain learning and multi-domain learning in the results of unseen target evaluation (Out Avg. in Table 6), it is evident that the performance with multiple datasets is significantly superior. Effectively addressing conflict and discrepancy issues, and learning from diverse and complex situations across multiple domains, leads to the development of a general model with better performance on unseen domains.

#### 4.2.3 Low-shot adaptation results (E3)

In E3, we delve into low-shot learning, examining how well the model trained with multiple sources in the leave-one-out experiments (E2) adapts to unseen targets. In this experiment, we utilized 10% of the target training set for low-shot learning (in a 3-fold setting). For datasets with abnormal categories, an equal number of data per category are randomly sampled, while for uncategorized datasets like UBIF and ST, random sampling is employed.

**Importance of general pre-trained baselines for adaptation.** In Table 7, comparing with Table 6, notable performance improvements are observed across all baselines for target samples with severe domain conflicts and gaps, such as UBIF, TAD, and ST, even with a limited volume of training samples. This emphasizes the importance of building a general model to adapt on unforeseen targets. While the MIL baseline with a single head holds an advantage in adapting specific single head when target samples are appropriately selected, it may become dependent on specific samples and is susceptible to overfitting. On the other hand, NullAng-MIL, with conflict-aware learning and training on diverse domains, outperforms other methods in adaptation performance.

#### 4.2.4 Full fine-tuning results (E4)

**Multi-domain models superior to specific single-domain models.** Table 8 shows the full fine-tuning results of single-domain and multi-domain trained baselines. The single-domain models reflect results from training and testing within a specific dataset. When evaluating multi-domain baselines trained on all datasets in the held-in (E1) setting, it demonstrates performance comparable to single-domain models that perform well by fitting to a single dataset. However, after full fine-tuning on the target dataset, the E1 model achieves superior performance in most cases. The E1

Table 8: **E4**: Comparison between the single-domain model and full fine-tuned models from the E1 and E2.

| MDVAD Benchmarks | | | | | | |
|---|---|---|---|---|---|---|
| | Target | | | | | |
| | UCFC | XD | LAD | UBIF | TAD | ST |
| *Single-domain* | | | | | | |
| Single | 77.93 | 83.23 | 83.82 | 92.62 | 90.75 | 90.79 |
| *E4: Full fine-tuning* | | | | | | |
| E1 | 78.62 | 82.71 | **84.41** | 92.42 | **92.50** | 91.17 |
| E2 | **80.24** | **82.77** | 83.81 | **92.95** | 92.07 | **91.27** |

**Finetuning:** Target dataset / **Testing:** Target dataset

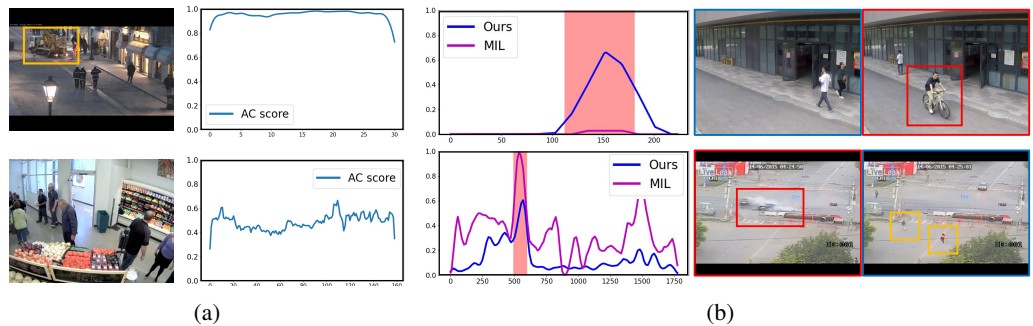

(a)                                                                    (b)

Figure 3: **(a)** The plot of AC scores. Both scenes are from UCFC and are normal in UCFC. (Top) Yellow box indicates abnormal conflict, which is abnormal in ST. (Bottom) Normal scene. **(b)** Qualitative results. Red box indicates abnormal event in the scene. (Top) *Bicyclist on walkway* abnormal event in ST. (Bottom) *Accident* abnormal event in UCFC and *Pedestrian on Road* abnormal conflict in TAD.

and E2 models, which are well-explored across multiple domains, outperform the models specifically trained on a single domain after full fine-tuning.

#### 4.2.5 Summary

**Null-MIL and NullAng-MIL.** The Null-MIL baseline achieves the best performance (86.86%) in the Held-in setting (E1), even surpassing the in-domain average results (86.52%). However, in the Leave-one-out setting (E2), which involves unknown target domains, it shows suboptimal performance. This is because it is impossible to determine which head's output score among the source dataset heads is more reliable, so multiple head baselines employ the maximum score for unknown domains. Conversely, NullAng-MIL exhibits superior results in E2 due to its similarity-based score calculation. NullAng-MIL considers the cosine distance between the final feature and the weights of each source head. As a result, when calculating the final score, the head with high similarity to the target data is activated, yielding promising results. When pre-training the general VAD model without knowledge of the target, we observed that utilizing the NullAng-MIL baseline is beneficial. However, when conducting multi-domain learning with knowledge of the target, employing Null-MIL yields better results. We observed that when pre-training the general VAD model without prior knowledge of the target, utilizing the NullAng-MIL baseline proves beneficial and employing Null-MIL yields better results when knowledge of the target is available during multi-domain learning.

**Role of the AC Classifier.** In multi-domain learning, the proposed AC classifier serves as an auxiliary branch designed to predict whether there is an abnormal conflict between domains, thus making the model aware of conflicts. Since it is not used during testing, it does not impact the final model's cost but provides performance gains in most experiments. Notably, in E2 and E3, training with the AC classifier shows a significant boost effect, aiding in adaptation to unseen domains.

### 4.3 Discussions

**Open-set VAD.** We conducted experiments in an open-set scenario using UBNormal (UBN) dataset [1]. The UBN is a VAD benchmark proposed for open-set scenarios to handle unexpected abnormal events. Both normal and abnormal events are available during training, but the anomalies that occur during inference belong to a distinct set of anomaly types (categories). Unlike other VAD datasets, UBN consists of synthetic videos to alleviate the difficulty of collecting abnormal event data in the real world. There are substantial abnormal conflicts and differences in the visual settings of scenes compared to other domains. For multi-domain learning, we reorganized the training/ testing set of UBN to maintain the same number of normal and abnormal videos in the training set with MDVAD, which is uniformly sampled and balanced the same amount of videos from all domains. As

Table 9: Ablation studies on Open-set VAD scenario

| Single source: UBN / Target: UBN | |
|---|---|
| MIL | 75.13 |
| **Multiple source: MDVAD + UBN / Target: UBN** | |
| MIL | 67.95 |
| MIL+AC | 70.56 |
| Null-MIL | 72.14 |
| Null-MIL+AC | 70.94 |
| NullAng | 74.42 |
| NullAng+AC | 74.54 |

shown in Table 9, despite domain discrepancies and abnormal conflict, the model effectively handles multiple domain learning, demonstrating that general feature learning can adequately address unseen abnormal categories. Please refer to §. A5 for more details.

**Comparison with WVAD models.** Although this paper focuses on the analysis of multiple domain learning within the context of abnormal conflict issues, instead of exploring complex architecture designs for single-domain VAD models, we compare with various VAD models. Table 10 presents the results of other MIL-based WVAD models, MMIL [43], ARNet [46], WSAL [30], and COMO [7] on the MDVAD task. Compared to the proposed baseline trained with the AC classifier, our method achieves the highest average AUC, particularly in datasets with severe abnormal conflicts and scene discrepancies, such as TAD and ST, in both the E2 and even more in the E3 settings. Various single-domain VAD models or backbones can be incorporated into the MDVAD task, showing a direction for future generalization work. Please refer to § A6.

Table 10: WVAD models on MDVAD Benchmark

| Models | Target | | | | | | |
|--------|--------|----|-----|------|-----|----|------|
| | UCFC | XD | LAD | UBIF | TAD | ST | Avg. |
| *E1: Held-in* | | | | | | | |
| MMIL | 77.93 | 81.34 | 85.18 | 85.44 | 87.78 | 84.39 | 83.68 |
| ARNet | 79.26 | 80.38 | 85.27 | 84.18 | 89.57 | 86.65 | 84.22 |
| WSAL | 76.47 | 78.35 | 75.44 | 86.41 | 85.62 | 82.44 | 80.79 |
| COMO | 80.41 | 82.75 | 86.24 | 85.82 | 90.13 | 89.76 | 85.85 |
| **Ours** | 77.21 | 82.09 | 83.88 | 91.90 | 91.36 | 91.12 | **86.26** |
| *E2: Leave-one-out* | | | | | | | |
| MMIL | 76.68 | 74.92 | 67.39 | 82.40 | 67.61 | 61.86 | 71.81 |
| ARNet | 77.05 | 75.02 | 78.98 | 80.84 | 75.09 | 55.34 | 73.72 |
| WSAL | 76.59 | 73.75 | 76.71 | 79.86 | 77.68 | 55.54 | 73.36 |
| COMO | 77.07 | 76.64 | 77.43 | 76.74 | 78.67 | 57.63 | 74.92 |
| **Ours** | 78.55 | 77.68 | 77.36 | 82.53 | 79.21 | 60.41 | **75.96** |
| *E3: Low-shot Adaptation* | | | | | | | |
| MMIL | 77.28 | 72.75 | 80.70 | 85.29 | 81.72 | 61.33 | 76.51 |
| ARNet | 75.48 | 72.18 | 79.90 | 81.70 | 79.43 | 68.24 | 76.16 |
| WSAL | 75.77 | 72.18 | 66.45 | 82.57 | 75.05 | 78.05 | 75.01 |
| COMO | 70.02 | 72.89 | 80.59 | 83.05 | 74.73 | 73.53 | 75.80 |
| **Ours** | 78.99 | 75.80 | 77.82 | 85.75 | 84.06 | 76.23 | **79.78** |
| *E4: Full Fine-tuning* | | | | | | | |
| MMIL | 80.26 | 82.51 | 86.54 | 89.88 | 90.32 | 89.34 | 86.48 |
| ARNet | 80.88 | 82.57 | 86.72 | 89.92 | 90.69 | 91.7 | 87.08 |
| WSAL | 80.99 | 81.6 | 76.43 | 87.87 | 88.78 | 87.47 | 83.86 |
| COMO | 80.61 | 84.25 | 86.88 | 91.51 | 91.74 | 91.23 | **87.70** |
| Ours | 78.62 | 82.71 | 84.41 | 94.42 | 92.5 | 91.17 | 87.31 |

**AC classifier.** The AC classifier helps the model learn conflict-aware features, providing a clearer understanding of abnormalities. The proposed framework is composed of Domain-Agnostic Layers that learn general features across multiple domains, followed by Multiple Heads that predict abnormalities specific to each domain. When the Domain-Agnostic layers learn to perform AC classification, they capture whether the input snippets relate to abnormal conflict or not. From the Heads' perspective, these features are separated into abnormal conflict and non-AC in the feature space, allowing the Heads to apply different criteria (decision boundaries) when distinguishing between normal and abnormal instances. For example, in classifying abnormalities, non-AC scenarios can be addressed more straightforwardly, while abnormal conflict scenarios require a more careful exploration. Fig. 3(a) presents the results of the AC classifier of NullAng-MIL baseline from E1. It shows abnormal conflict scores for two scenarios: (Top) *A car on the Sidewalk* abnormal event in ST, which is normal in the UCFC. (Bottom) A normal situation of people shopping in a grocery store. Through the multi-domain learning, the AC classifier outputs a high abnormal conflict score for the top sample, demonstrating that the model has learned to be conflict-aware.

**Qualitative results.** Fig. 3(b) illustrates abnormal conflict scene: (Top) *Bicyclist on Walkway*, abnormal in ST but normal in other domains, and (Bottom) *Pedestrian on Road*, abnormal in TAD but normal in UCFC. In these scenarios, the MIL baseline trained with multiple domains predicts (Top) false negative and (Bottom) false positive, while ours adaptively handles conflicts across different domains.

## 5 Conclusion, Limitation, and Future Works

In this paper, we propose a new task called MDVAD, whose ultimate goal is to effectively learn from multiple domains with different data distributions and definitions of abnormality without confusion, resulting in a general VAD model. As a baseline, we propose a new multi-head framework with Null(Ang)-MIL loss and AC classifier. These modules effectively handle abnormal conflicts between domains and show meaningful results in the MDVAD benchmark with diverse evaluation protocols.

Instead of exploring complex architecture design of single-domain VAD models, this paper focuses on resolving abnormal conflicts from multiple domains. Various single-domain VAD models or backbones can be applied into our novel framework to address the MDVAD task, representing a valuable direction for future generalization research.

## Acknowledgments

This work was partly supported by the NAVER Cloud Corporation and the National Research Foundation of Korea (NRF) grant funded by the Korea government(MSIT)(RS-2024-00456589).

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

# Supplementary Material

Table A1: Detailed descriptions of VAD datasets used in the paper. N: The number of normal videos. A: The number of abnormal videos.

| Datasets | Videos | Frames | Train / Test Volume | | Anomalies (Categories) | Settings | Sample |
|---|---|---|---|---|---|---|---|
| UCFC | 1900 (128h) N: 950 / A: 950 | 7247 per video | Train: 1610 N: 800 / A: 810 | Test: 290 N: 150 / A: 140 | Abuse, Arrest, Arson, Assault, Accident, Burglary, Explosion, Fighting, Robbery, Shooting, Stealing, Shoplifting, Vandalism | CCTV |  |
| XD | 4754 (217h) N: 2349 / A: 2405 | - | Train: 3954 N: 2049 / A: 1905 | Test: 800 N: 300 / A: 500 | Abuse, Car Accident, Explosion, Fighting, Riot, Shooting | CCTV, Sports, Games, Movies, News |  |
| LAD | 2000 N: 1238 / A: 762 | - | Train: 1440 N: 958 / A: 482 | Test: 560 N: 280 / A: 280 | Crash, Crowd, Destroy, Drop, Falling, Fall Into Water, Fighting, Fire, Hurt, Loitering, Panic, Thiefing, Trampled, Violence | CCTV |  |
| UBIF | 1000 N: 216 / A: 784 | - | Train: 933 N: 757 / A: 176 | Test: 67 N: 27 / A: 40 | Fights | CCTV, Mobile |  |
| TAD | 500 (25h) N: 250 / A: 250 | 1075 per video | Train: 400 N: 210 / A: 190 | Test: 100 N: 40 / A: 60 | Vehicle Accidents, Illegal Turns, Illegal Occupations, Retrograde Motion, Pedestrian on Road, Road Spills, The Else | Traffic CCTV |  |
| ST | 437 N: 330 / A: 107 | 726 per video | Train: 238 N: 175 / A: 63 | Test: 199 N: 155 / A: 44 | Fighting, Robbery, Chasing, Jumping, Throwing, Throwing, Running, Dropping, Motorcycle, Skateboard, Car, Gun, Loitering | Campus CCTV |  |
| CADP | 1416 N: 0 / A: 1416 | - | Train: 1416 N: 0 / A: 1416 | - | Traffic accident | Traffic CCTV |  |
| NWPU | 547 N: 423 / A: 124 | - | Train: 305 N: 305 / A: 0 | Test: 242 N: 118 / A: 124 | Group conflict, Trucks, Climbing fence, Protest, Cycling on footpath, Dogs, Kicking trash can, Chasing, Loitering, Car crossing square, Scuffle, Littering, U-turn, Falling, etc. | Campus CCTV |  |

**Summary** The supplementary material is organized in the following order. First, it provides an explanation of the characteristics of the Video Anomaly Detection (VAD) dataset and introduces the detailed configuration and experimental settings of the MDVAD benchmark. Following this, we present implementation details of the experiments and examples and explanations related to abnormal conflict. Lastly, we discuss the pseudo labeling equation, score comparison plots, and failure cases. Note that back-references in the supplementary material sections and tables are from the main manuscript.

**Terminology**

- **Domain/ Dataset:** In this paper, 'domain' refers to the broader context or environment, while 'dataset' refers to the specific collection of data used within that domain.

- **Single-domain:** Training and evaluation data are the same, as in traditional VAD research.

- **Multi-domain Learning/ General VAD:** General VAD is the objective of detecting anomalies across various domains, while Multi-domain Learning is the method used to achieve this by training on multiple domains.

- **Multi-head Learning:** The baseline method proposed in this paper for Multi-domain Learning.

- **Multi-task Learning:** Performing multiple tasks with a single model, related to E1 by integrating various domain's anomalies into one framework.

- **General Pre-training:** A methodology of pre-training on large data without target knowledge, related to E2 by training in a held-out setting and applying to an unseen target. Furthermore, general pre-trained models adapt well to low-shot learning in E3 and fine-tuning in E4.

## A   VAD Datasets

As mentioned in §2, the VAD task encompasses diverse datasets, as delineated in Table A1.

UCFC [43], the prominent large-scale dataset for the Weakly-Supervised approach, consists of untrimmed surveillance videos. It categorizes anomalies into 13 classes related to public safety,

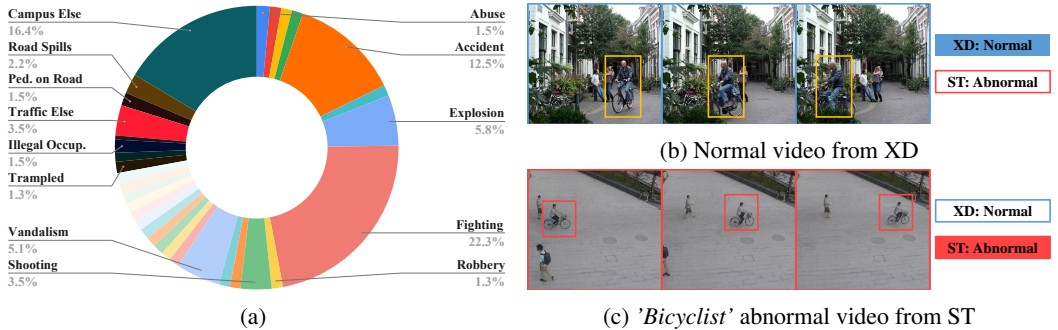

(a)

(b) Normal video from XD

(c) *'Bicyclist'* abnormal video from ST

Figure A4: **(a)** Distribution of the training videos by abnormal category in the MDVAD benchmark. **(b)(c)** Examples of abnormal conflict between VAD datasets. The scenes colored by red and blue borders represent abnormal and normal situations, respectively, based on the labels from the corresponding dataset.

ranging from crimes such as abuse and arson to more complicated scenarios like stealing and shoplifting. The dataset consists of a total of 950 normal and abnormal videos each, with the training set maintaining a balanced distribution of 800 and 810 samples for normal and abnormal videos, respectively.

The XD dataset [51] stands out as the most extensive dataset with a variety of environments, including CCTV, movies, hand-held cameras, and car camera settings. It incorporates six categories of violence, such as abuse, riot, and explosion, defining anomalies. Within the dataset of 4,754 videos, there are 2,049 normal and 1,905 abnormal training videos.

The LAD dataset [47] is defined by a detailed categorization of 14 anomaly classes, including crash, fire, and violence, consisting of 2,000 videos, with 958 normal and 482 abnormal videos.

The UBIF dataset [9] is composed of various fighting-related anomalies extracted from our daily life videos. This dataset consists of 1,000 videos without a specific categorization for the fighting class, and the training set consists of 757 normal and 176 abnormal videos, leading class imbalance issue.

The TAD dataset [19] is structured with traffic surveillance and 1st-person videos, encompassing anomalies related to traffic, such as accidents, illegal turns, and road spills. This dataset, totaling 500 videos, has a small volume with 210 and 190 numbers of normal and abnormal training videos.

The ST [24] serves as a campus surveillance dataset, capturing anomalies occurring on pedestrian walkways. It exhibits a heightened sensitivity to abnormal events, ranging from fighting and throwing to running and jumping. With only 175 normal and 63 abnormal videos in the training set out of a total of 437 videos, there is a notable imbalance between classes, and the quantity of data is insufficient for learning. Abnormal categories in the training set cover all anomalies in the testing set on each dataset except for ST (ST remains indeterminate because of the absence of category information), which means evaluations are exclusively conducted for seen abnormal events.

Examining the dataset samples in Table A1 reveals that TAD and ST exhibit a visual gap compared to the other datasets, UCFC, XD, and LAD. Such gap is attributed to visual elements such as scene settings and conditions that impact the visual characteristics of scenes, which is the scene discrepancy discussed in §2. While LAD (some videos) and ST are recorded in square or campus settings to establish real-world scenario datasets, the remaining datasets are constructed online, such as YouTube and LiveLeak using text search queries. These datasets also exhibit a significant domain distance, as indicated in Table 3. As such, the criteria for defining abnormalities and the visual characteristics of scenes vary across different datasets.

## B  MDVAD Benchmarks

### B.1  Configuration

The five datasets introduced in Table A1 have different training set volumes, leading to challenges in handling multiple dataset learning, where larger datasets may overwhelm the learning process.

For instance, there is a significant disparity in training set volumes, with TAD (400 videos), ST (238 videos), and XD (3954 videos) differing by almost 10 times. Additionally, the number of abnormal videos in XD differs roughly 10 and 30 times compared to TAD and ST, respectively.

The objective of the MDVAD task is to avoid abnormal conflict and build a general model through balanced learning across diverse situations and anomalies. To control for other variables, we uniformly *sampled* and *balanced* the same amount of videos from all domains, equal to the number of videos in the smallest domain. Since TAD and ST have an insufficient number of videos for training, we recombine them with two new datasets, CADP [41] and NWPU [3], respectively, to balance the numbers. CADP is a dataset for traffic accident analysis, and it shares similarities with TAD in terms of setting and abnormal definition. Hence, following the approach in [28], we compose TAD's training set by adding abnormal videos from CADP. Additionally, incorporating the recently released NWPU dataset, which collects various anomalies of pedestrians on campus, we augment ST's dataset with NWPU data. For both datasets, we conducted random sampling for normal and abnormal videos, considering the smallest sample size, resulting in a number of 210 normal and 176 abnormal videos. In this process, for datasets with information on abnormal classes, we compose to achieve a uniform number of instances per abnormal category on each dataset. For datasets with a sufficient amount, we organize 3-fold sets to minimize overlap. All reported performance metrics are the average results of evaluations conducted on the target dataset after training on each fold. For example, in the E2 leave-one-out setting, the average results are derived from models trained on each fold and evaluated on an E3 low-shot adaptation set for the corresponding fold.

The MDVAD benchmark incorporates the datasets introduced in Table A1, including abnormal categories: *Abuse, Arrest, Arson, Assault, Accident, Burglary, Explosion, Fighting, Robbery, Shooting, Stealing, Shoplifting, Vandalism, Drop, Hurt, Fall Into Water, Falling, Destroy, Fire, Violence, Crowd, Thiefing, Panic, Loitering, Trampled, Illegal Turns, Illegal Occupations, Retrograde Motion, Traffic Else, Pedestrian on Road, Road Spills, and Campus Else*. The circular chart in Fig. A4(a) illustrates the distribution of categories. For datasets without abnormal categories like ST (named as *Campus Else*), TAD (where additional categories are labeled as *Traffic Else*), and UBIF (where categorization is not specified and all fall under *Fighting*), we provide appropriate category names. The averages across the 3-folds are calculated, and in the case of XD with multiple labels per video, we count the number of categories per video. The *Fighting* predominates in the majority of datasets, occupying the largest portion, followed by categories like *Accident*, *Explosion*, and others.

## B.2    Experimental settings

VAD is predominantly evaluated using the Area Under the Curve (AUC) score based on abnormal scores, focusing on the trade-off between sensitivity and specificity, emphasizing the false positive rate (FPR). While XD employs the Average Precision (AP) metric to balance precision and recall and concentrate on positive samples for direct comparison in cross-dataset and held-in/out evaluations, and emphasizing False Positive Rate (FPR) (detailed explanation is in Section D), we utilize the AUC score across all evaluations.

The MDVAD benchmark comprises four protocols: Held-in (E1) involves multiple-source learning with all datasets, Leave-one-out (E2) employs a leave-one-out approach where a dataset left out during training is used for evaluation, low-shot adaptation learning (E3) is training on the remaining datasets, and Full fine-tuning (E4) that fine-tunes a multi-domain models on single-domain datasets. In the E3 setting, around 10% of the training set, 20 videos, is randomly sampled for low-shot examples. For datasets with anomaly category information, low-shot examples are selected uniformly to ensure an even distribution within each category. To mitigate the impact of randomness, the low-shot setting is also conducted with 3-fold cross-validation, obtaining low-shot examples from each fold. In all cases, including MIL, Null-MIL, and NullAng-MIL models, all weights except for the pretrained backbone are updated. Additionally, for abnormal videos, the score from the AC classifier is multiplied to the loss value.

## C    Implementation Details

All experiments are conducted on the MDVAD benchmark. We use I3D backbone feature which is pretrained on Kinetics dataset [4] with $C = 2048$ dimension of RGB features. During the training phase, we conduct consistent experiments with a batch size of 32, utilizing the Adam optimizer [20]

with a learning rate of $5e-5$ and weight decay of $5e-4$. The input image size is set to $224 \times 224$ and following [21, 31, 45, 7], we perform 10-crop augmentation when extracting backbone features. The backbone is used the same as RTFM [45] and CoMo [7]. For input data, 16 frames are stacked for a snippet, and $T$ snippets are uniformly sampled in the video during training. For testing, all snippets pass through the model, and the output score is assigned to all frames within the snippet. For MIL learning, the Top-$K$ snippets, where $K$ equals 10% of the total, *i.e.*, $K = 3$, are utilized. In the case of a single-headed MIL model in Table 5 (E1) setting, since pseudo-labels cannot be generated through multiple heads, the AC classifier is trained with a label that $y^{AC} = 1$ for a score $s_{D_1}^a$ within the $[0.4, 0.7]$ range. When computing EMD metric in §2, we use a pretrained I3D backbone model for extract embedding features of each dataset and measure the distance between feature vectors. We use a test set with frame-level annotations for abnormalities. All models and experiments are implemented and evaluated end-to-end using PyTorch [34] with a single NVIDIA V100 GPU.

In weakly-supervised VAD, maintaining class balance is pivotal, so the number of normal and abnormal input videos within a batch is kept equal for the MIL-based learning. Similarly, during multiple source learning, an equal number of input videos from each source domain within a batch is crucial. Despite each head is trained independently, the final feature $\mathbf{F}_{DA}$ is extracted from a shared domain-agnostic part, necessitating the balance in the number of source domains. Moreover, when training the AC classifier, in order to address the imbalance between the number of abnormal conflict snippets and all normal/abnormal snippets, we apply focal loss [23] to $L_{AC}$. During the testing phase, the AC classifier is eliminated, and the output is calculated for each input snippet $\mathbf{v}_i$ as the final Abnormal Score$_i$.

Table A4 provides information about the layers of the baseline model. `Conv1d`, `fc`, `Max`, `BN`, and `DO` represent 1D convolutional layer, fully connected layer, element-wise max activation, 1D batch normalization [16], and dropout [42], respectively, with a dropout probability set to $p = 0.7$. In `Conv1d`$(c, k, s)$ and `fc`$(c)$, $c$, $k$, and $s$ denote the channel size, kernel size, and stride size, respectively. $T$ input snippets pass through the backbone, resulting in a backbone feature $B$ with a shape of $(N, T, C)$, where $N$ represents the batch size. The feature aggregation layer undergoes channel doubling followed by channel squeezing with element-wise max operation, resulting in an output feature $\mathbf{F}_{agg}$ with a shape of $(N, T, C)$. Subsequently, the temporal aggregation layer outputs $\mathbf{F}_{DA}$ with a shape of $(N, T, C/2)$. These embeddings are then input into the AC classifier and `fc` layer. The AC classifier, composed of consecutive `fc` layers, outputs AC scores $\mathbf{s}^{AC}$ for each snippet with a shape of $(N, T, 1)$. Additionally, the final feature $\mathbf{F}$, processed through an `fc` layer, is input into multiple domain heads $\mathbf{w}_{D_d}$ for NullAng-MIL learning. During this process, both the feature and weights are normalized with respect to $\|\mathbf{F}\|$ and $\|\mathbf{w}_{D_d}\|$ for each domain.

# D  Abnormal Conflict

In Fig. A4, (a) represents a video labeled as normal in UCFC, while (c) is labeled as abnormal in TAD, both depicting a situation where people are present on the road. Similarly, (b) is labeled as a normal video in XD, whereas (d) is considered abnormal in ST; both scenes show bicycling on a

Table A2: Abnormal Conflict: Average of Relative FPR and Relative FNR.

| Source | Target | | | | | |
|---|---|---|---|---|---|---|
| | UCFC | XD | LAD | UBIF | TAD | ST |
| UCFC | - | 0.204 | 0.099 | 0.153 | 0.195 | 0.273 |
| XD | 0.115 | - | 0.082 | 0.234 | 0.166 | 0.326 |
| LAD | 0.186 | 0.152 | - | 0.275 | 0.175 | 0.337 |
| UBIF | 0.074 | 0.147 | 0.150 | - | 0.236 | 0.286 |
| TAD | 0.237 | 0.340 | 0.281 | 0.266 | - | 0.343 |
| ST | 0.219 | 0.305 | 0.265 | 0.288 | 0.354 | - |

Table A3: Ablation of Eq. 6 in manuscript.

| | Target | | | | | || Avg. |
|---|---|---|---|---|---|---|---|
| | UCFC | XD | LAD | UBIF | TAD | ST || |
| *E1*: Held-in results | | | | | | || |
| Fixed | 76.50 | 81.85 | 82.40 | 91.56 | 91.59 | 91.02 || 85.82 |
| Std. | 77.04 | 82.28 | 83.04 | 91.71 | 91.61 | 90.75 || 86.07 |
| Diff. | 77.21 | 82.09 | 83.88 | 91.90 | 91.36 | 91.12 || 86.26 |

Table A4: Illustration of each layer of proposed model.

| Part | Name, Notation | | Layers | Output |
|---|---|---|---|---|
| Domain Agnostic | Backbone | **B** | **I3D** | $(N, T, C)$ |
| | Feature Aggregation | $\mathbf{F}_{agg}$ | `Conv1d`$(2C, 3, 1)$ `Max`-`ReLU`-`BN` | $(N, T, C)$ |
| | Temporal Aggregation | $\mathbf{F}_{DA}$ | `Conv1d`$(C/2, 3, 1)$ -`ReLU`-`BN` | $(N, T, C/2)$ |
| | AC Classifier (Train-Only) | $\mathbf{s}^{AC}$ | `fc`(512)-`ReLU`-`DO` `fc`(128)-`ReLU`-`DO` `fc`(1)-`Sigmoid` | $(N, T, 1)$ |
| | FC layer | **F** | `fc`(128)-`ReLU` | $(N, T, 128)$ |
| Domain Specific | Multiple Heads | $\mathbf{w}_{D_d}$ | `fc`($D$) | $(N, T, 2D)$ |

walkway. We define such instances, where the criteria for abnormalities differ across datasets, as *abnormal conflicts*.

While cross-domain evaluation in Table 2 reveals a domain gap, it is difficult to attribute as abnormal conflicts directly. In Table A2, we calculate the average of the relative False Positive Rate (FPR) and relative False Negative Rate (FNR) to quantify the abnormal conflict between domains.

$$AC_{i,j} = \frac{(FPR_{i,j} - FPR_{j,j}) + (FNR_{i,j} - FNR_{j,j})}{2},$$ (8)

where $i$ and $j$ indicate source and target domain, repectively. First, FPR and FNR refer to the rate of misclassifying abnormal as normal and normal as abnormal in the scenario of learning from a source and testing on a target, respectively. However, since these include the FPR and FNR caused by the incompleteness of the model itself, this cannot accurately reflect the abnormal conflict between the source and target dataset. Therefore, we subtract the FPR and FNR when the source domain is the same as the target domain ($FPR_{j,j}$ and $FNR_{j,j}$) from the cross-domain FPR and FNR to calculate the relative FPR and FNR caused by the domain shift. Table A2 shows similar trends compared to Table 2, where values are not normalized. This shows that abnormal conflicts between domains are relevant to the reduction in generalization ability that can occur when transferring between domains.

## E   Related Works

As mentioned, proposing a new VAD model is beyond the scope of this paper, this part is organized to aid understanding.

**Unsupervised VAD**   Defining anomalies precisely is challenging as they can vary based on criteria and sensitivity, and it is impossible to categorize and collect datasets for every scenario. For this reason, Unsupervised VAD methods [33, 5, 24, 15] have been developed to learn normal patterns using only normal training data and then detect anomalies as scenes with low normality during the test phase. However, this approach leads to a significant bias towards the normal samples, causing the detector to misclassify normal patterns that differ from the learned data as abnormal events, resulting in a high false alarm rate. To address this issue, Weakly-supervised VAD (WVAD) approaches [55, 44, 43, 50, 21, 7] have been introduced to differentiate between normal and abnormal events with minimal supervision using video-level annotations, achieving significant performance improvements while incurring lower labeling costs compared to frame-level annotations.

**Weakly-supervised VAD**   MMIL [43] is the first WVAD method to pose the MIL ranking approach as a regression problem that achieves significant performance gains. Various learning methods, including magnitude feature learning [45], self-training approaches through pseudo labels [13, 21], and distance learning [53], have been explored alongside traditional MIL methods. Furthermore, subsequent approaches have focused on capturing features by focusing temporal or motion information [58], leveraging additional audio signals [51], learning relations between temporal [50] or motion and context information [7]. These methods differentiate complex anomalies through contextual information; for example, the same motion may be classified as normal or abnormal depending on the surrounding environment. PFMP [26] proposed a method to utilize virtual data anomalies to reduce scene discrepancy with real-world data, addressing the issue of data scarcity in VAD. However, these approaches still cannot address abnormal conflicts, where the same situation is labeled differently as abnormal or normal based on the criteria of different datasets.

**Generalization for VAD**   A few studies have been conducted to generalize VAD. *Domain adaptation* refers to evaluating a new domain that was not seen during training, and some works have proposed using some data from the target domain [27, 29] or not using it at all [2]. We also experiment with domain adaptation, but the difference is that we use multiple domains compared to existing studies that only use a single domain during training. *Open-set recognition* aims to work well even in untrained classes (abnormal categories in VAD). Compared to our research involving multiple domains, Open-set VAD [59, 1] focuses on learning and evaluation within a single domain.

**Multi-domain learning**   Multi-Domain Learning (MDL) refers to a method for learning datasets from multiple domains with various distributions together. MDL originated from natural language processing [11, 17] and has been applied to various tasks in computer vision [36, 37, 22, 32, 49,

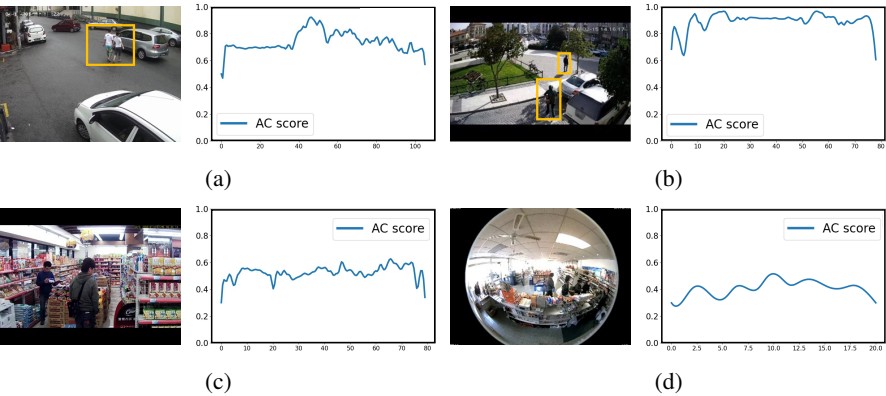

Figure A5: The plot illustrates the abnormal conflict scores from the AC Classifier on normal videos on UCFC. For clarity, all scores have been normalized using the minimum and maximum values of the entire score distribution produced by the AC Classifier.

48, 56, 18]. The difficulty of MDL arises when the difference in distribution between domains is significant, and learning domain-invariant features is the basic approach [39]. In the fields of image classification [36, 37, 22], object tracking [32], detection [49], and segmentation [48], methods for learning multiple domains with multiple heads have been proposed. After that, methods have been proposed to have a unified label space [56, 18], but in this process, the problem that the labels are different for each domain arises. [18] solved the label conflict problem by proposing class-independent loss using the *Null* class strategy. On the other hand, VAD has only two classes, normal and abnormal, so there is no explicit label conflict problem, but there is an implicit abnormal conflict problem where the definition of a label is different for each domain.

A similar method is multi-task learning [54], which considers differences between tasks rather than domains. In addition, multi-source domain adaptation [35] utilizes data from the target domain, and domain generalization [10, 40, 12] requires adaptation to an unseen target domain. The difference with these tasks is that they basically share the label space between the source and target domains, while MDL for VAD has different label definitions.

## F Discussions

### F.1 Pseudo label of AC Classifier

The purposes of the AC Classifier is the facilitation of feature learning with consideration of the discrepancies across multiple domains. In cases where snippet is entirely normal or abnormal across all datasets, the AC score is low, making it difficult to directly use it for the abnormal score. Noisy samples can indeed have a negative impact on training in Eq. 6. However, in cases where even one of the multiple datasets has a different definition of normal and abnormal, it is crucial to sensitively detect conflicts in such samples, leading us to decide that using a difference is preferable. In Table A3, we conducted ablation studies with Std. ($\tau = 0.1$) and a fixed value ($y_i^{AC} = 1$ where $0.3 < s_{D_d,i}^a < 0.7$) which revealed no significant differences between the methods.

### F.2 Abnormal conflict score

Fig. A5 depicts the output abnormal conflict scores from the AC classifier of the E1 held-in model. While all events are considered normal in UCFC, (a) and (b) correspond to the *Pedestrian on Road* abnormal category in TAD (as shown in Fig. 1(c)). Therefore, unlike (c) and (d), the normal situations of people shopping in a grocery store, (a) and (b) exhibit elevated abnormal conflict scores.

### F.3 Abnormal score

Fig. A6 compares the abnormal scores of our model, trained with NullAng-MIL + AC classifier, and the MIL model trained with MIL loss on a single head in the E2 multiple dataset setting. In scenario (a), before the explosion, an oil truck passes, causing the MIL model to generate a high abnormal

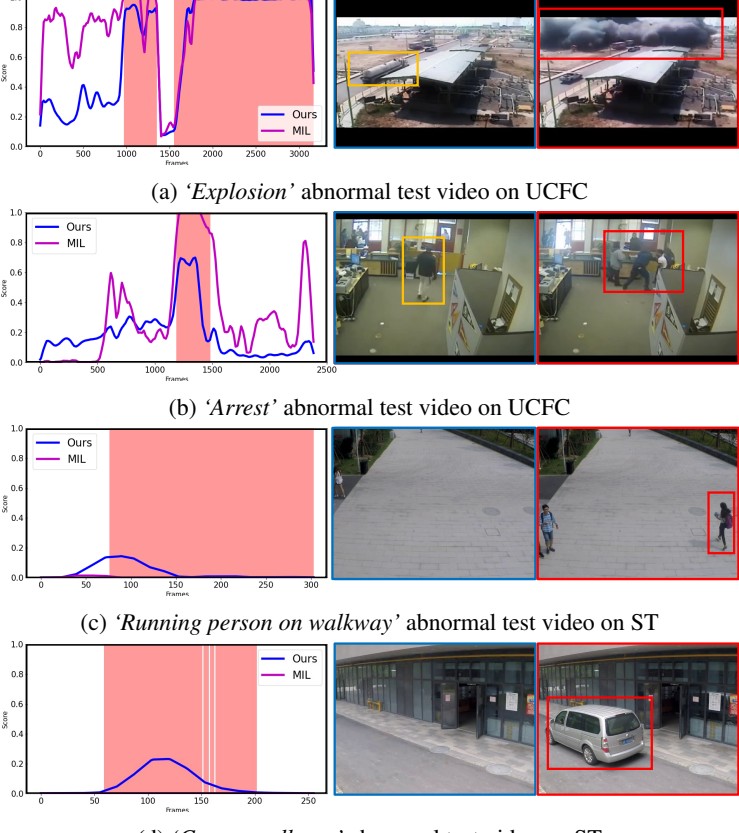

(a) *'Explosion'* abnormal test video on UCFC

(b) *'Arrest'* abnormal test video on UCFC

(c) *'Running person on walkway'* abnormal test video on ST

(d) *'Car on walkway'* abnormal test video on ST

Figure A6: The plot of abnormal scores with blue and magenta lines representing our model and the MIL baseline model (E1), respectively. The red region indicates the time when abnormal events occurred. The scenes are from input videos whose borders are colored red and blue for normal and abnormal scenes, respectively. For clarity, the scores from each model have been normalized.

score as a false alarm, while our model accurately predicts only the *explosion* abnormal situation. In scenario (b), where a person loitering the office, the MIL model outputs a high abnormal score, in contrast, our model that trained with multiple heads to avoid domain conflicts categorizes only the *arrest* defined in UCFC as an abnormal event.

Scenarios (c) and (d) involve abnormal videos in the ST dataset where a running person or car appears on the walkway. Because these scenarios are classified as normal scenes in other datasets during training, the MIL model considers them as normal. However, our model, recognizing them as abnormal scenes corresponding to ST, outputs high abnormal scores. Thus, when learning from multiple datasets, it is crucial to avoid conflicts by simultaneously learning domain-agnostic features with the AC classifier and predicting domain-specific scores using multiple heads according to each dataset's criteria.

## F.4 Multi-domain learning on Virtual dataset

The UBNormal (UBN) dataset [1] is a VAD benchmark proposed for open-set scenarios to handle unexpected abnormal events, where the abnormal categories in the train set and test set do not overlap. Unlike other VAD datasets which were collected from the real world, UBN consists of synthetic videos which leads

Table A5: Multi-source learning on MDVAD with UBN in E1

| Source | Target | | | | | | |
|---|---|---|---|---|---|---|---|
| **MDVAD+UBN** | **UCFC** | **XD** | **LAD** | **UBIF** | **TAD** | **ST** | **AVG.** |
| MIL | 78.59 | 81.79 | 85.06 | 86.6 | 88.83 | 87.99 | 84.81 |
| MIL+AC | 78.14 | 81.78 | 84.92 | 85.08 | 90.77 | 89.12 | 84.96 |
| NullMIL | 78.74 | 83.34 | 85.93 | 91.28 | 90.05 | 88.69 | 86.34 |
| NullMIL+AC | 78.76 | 83.19 | 86.01 | 92.63 | 90.73 | 90.54 | 86.98 |
| NullAngMIL | 77.09 | 81.01 | 83.96 | 92.88 | 91.57 | 90.04 | 86.09 |
| NullAngMIL+AC | 77.66 | 82.09 | 83.01 | 92.55 | 91.33 | 91.07 | 86.29 |

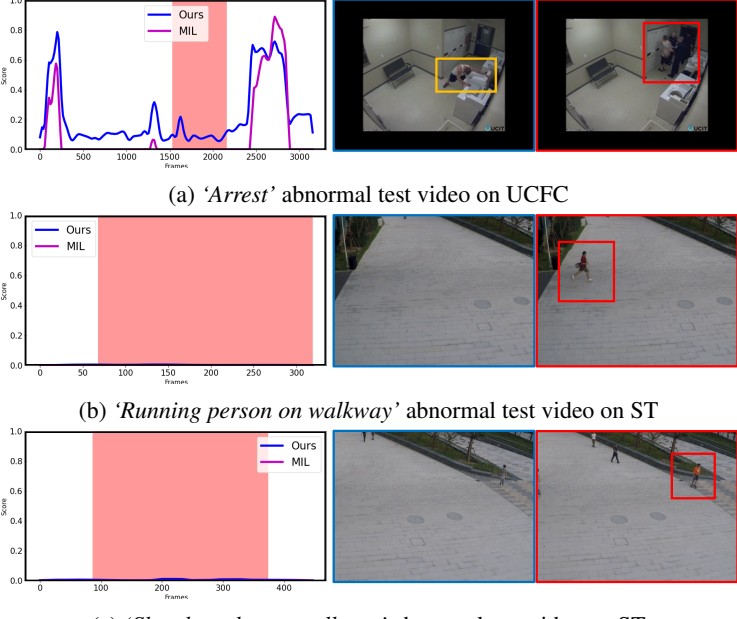

(a) *'Arrest'* abnormal test video on UCFC

(b) *'Running person on walkway'* abnormal test video on ST

(c) *'Skateboarder on walkway'* abnormal test video on ST

Figure A7: The abnormal scores of our model and the MIL model, where **(a)** corresponds to False Positive results, and **(b)(c)** represent False Negative results.

abnormal conflicts and differences in the visual settings of scenes. It comprises 543 videos across 29 scenes with 22 types of anomaly categories. Same as MDVAD benchmark, we reorganize Training set with 210 number of normal videos and 176 number of abnormal videos where training anomaly categories are *falling, dancing, walking injured, running injured, crawling and stumbling walk* and testing anomalies are *running, having a seizure, laying down, shuffling, walking drunk, people and car accident, car crash, jumping, fire, smoke, jaywalking and driving outside lane*.

The Table A5 shows the performance in each target domain when trained with MDVAD and UBNormal under the E1: held-in setting. In the single-head MIL baseline, there is a performance drop when trained with MDVAD+UBN, indicating difficulty in handling AC. However, the model trained with multiple heads and the AC classifier shows improved results. By leveraging the virtual dataset, we can overcome data limitations and create a general model capable of handling diverse and complex scenes.

### F.5   Baseline models

We conducted additional experiments using the WSAL [30] model as a baseline to validate our proposed method on the MDVAD benchmark's four protocols. The Table A6 presents the results, where all settings are consistent with the other experiments in the paper. The results show the addition of multi-head learning with NullAng-MIL and the AC Classifier to the WSAL model brings performance gains which effectively operate across

Table A6: Results of MDVAD with different baseline

| Exp. | Models | Target | | | | | | |
|------|--------|------|------|------|------|------|------|------|
| | | UCFC | XD | LAD | UBIF | TAD | ST | AVG. |
| E1 | WSAL | 76.47 | 78.35 | 75.44 | 86.41 | 85.62 | 82.44 | 80.79 |
| | +Ours | 76.90 | 78.59 | 76.17 | 87.83 | 81.89 | 86.27 | 81.27 |
| E2 | WSAL | 76.59 | 73.75 | 76.71 | 79.86 | 77.68 | 55.54 | 73.36 |
| | +Ours | 76.67 | 73.69 | 65.54 | 85.85 | 70.1 | 72.41 | 74.04 |
| E3 | WSAL | 75.77 | 72.18 | 66.45 | 82.57 | 75.05 | 78.05 | 75.01 |
| | +Ours | 76.81 | 72.38 | 59.51 | 83.49 | 77.6 | 74.57 | 74.06 |
| E4 | WSAL | 80.99 | 81.6 | 76.43 | 87.87 | 88.78 | 87.47 | 83.86 |
| | +Ours | 79.21 | 81.17 | 78.19 | 90.98 | 86.09 | 90.61 | 84.38 |

different baselines. Notably, in the E3 and E4, which shows the target adaptation of the pre-trained general model, the method yielded competitive results. Future studies can be delve deeper into studying more sophisticated baselines for resolving AC within the MDVAD setting.

### F.6 Failure cases

Fig. A7(a) presents an abnormal test video from UCFC corresponding to *Arrest* anomalies. However, this video includes scenes of *Stealing*, leading to the subsequent arrest and physical fighting. Both the MIL and our models output high abnormal scores for scenes other than the *Arrest*, as they correspond to *Stealing* and *Fighting* abnormal categories in UCFC. Models predict accurately in categories other than `Arrest`, but this failure is observed because of a single abnormal label for each video.

In Fig. A7(b), an abnormal event from ST is considered normal by both models, as other datasets are generally classified as normal. Similarly, in Fig. A7(c), the skateboarder is deemed normal, resulting in a false negative failure case. However, these situations are challenging to identify as severe abnormalities from a general perspective. This failure case is able to be addressed through low-shot learning or fine-tuning based on ST's criteria.

### F.7 Broader Impacts

The development of Multi-Domain Learning for Generalizable VAD enhances the reliability and accuracy of security and surveillance systems across diverse environments, thereby improving public safety. This research advances AI and machine learning by addressing domain adaptation challenges, promoting robust and generalizable systems. Additionally, it underscores the importance of ethical considerations and privacy, ensuring that advanced surveillance technologies are deployed responsibly and transparently.

