# OpenReview forum: "Towards Multi-Domain Learning for Generalizable Video Anomaly Detection"
_NeurIPS.cc/2024/Conference — NeurIPS 2024 poster_

### Official Review · Reviewer_926M · 2024-07-07

**Soundness:** 2
**Presentation:** 2
**Contribution:** 3
**Rating:** 5
**Confidence:** 4

**Summary:**

This work proposes a new task named Multi-Domain Learning Video Anomaly Detection, which aims to learn a general VAD model across domains. The work finds that abnormal conﬂict is a critical challenge in the task. Then, the work establishes a new benchmark, designs an effective baseline and conducts extensive experiments to investigate this challenge. The results shown on the benchmark demonstrate that the abnormal conﬂict is alleviated.

**Strengths:**

1. The work proposes a new task, which is interesting.
2. The work establishes a new benchmark to evaluate the new task.
3. The motivation of the proposed baseline, i.e., abnormal conﬂict, is clear and makes sense.

**Weaknesses:**

I have some concerns about the proposed method, and I think more comparison experiments are needed to demonstrate the effectiveness. Despite this, I think the abnormal conflict issue is interesting, thus I am willing to raise my rating if my major concerns are addressed. My concerns are as follows:

1. Why the proposed Abnormal Conﬂict (AC) classifer can address the abnormal conﬂict problem? Why the label is determined by the discrepancy in Eq. (6)? It seems that there are some mistakes in the formula (inconsistent with that in Fig. 2).
2. I would like to see the results of more baselines, in addition to MIL, Null-MIL and NullAng-MIL.
3. More detailed discussions about related works are needed, e.g., virual video anomaly detection datasets [1] and related techniques utilizing virtual datasets [2].

[1] Ubnormal: New benchmark for supervised open-set video anomaly detection, CVPR 2022

[2] Generating Anomalies for Video Anomaly Detection with Prompt-based Feature Mapping, CVPR 2023

**Questions:**

See the Weakness part.

**Limitations:**

The paper has discussed the limitations and potential impacts of the work.

---

> ### Author Rebuttal · Authors · 2024-08-07
>
> # [W1] AC Classifier
> Thank you for highlighting this important aspect.
>
> **Role of the AC Classifier:** By training with the AC Classifier, the domain-agnostic layer learns Conflict-Aware features, which helps in resolving conflicts.
>
> To achieve a general VAD model through multiple domain learning while avoiding abnormal conflicts, our framework aggregates features through domain-agnostic layers and performs multiple head learning. This activates only the output of the head corresponding to the input domain, assigning inactive heads with Null values to prevent confusion. While the heads are divided to prevent conflicts, the agnostic part extracts features from all datasets using a single branch. Therefore, to explore general features while being aware of abnormal conflicts, the AC classifier predicts conflicts by leveraging the variance in abnormal scores across multiple heads. This auxiliary task provides performance gains in most experiments. Notably, in E2 and E3 settings, it shows a significant boost effect, aiding in adaptation to unseen domains without any additional cost to the final model. Additionally, as shown in Supplementary Material Fig. A5, the conflict score of the AC classifier plotted for the UCFC domain reveals high conflict scores in scenarios like jaywalking, which is normal in UCFC but abnormal in other domains (TAD, ST), demonstrating the classifier's awareness of AC in the domain-agnostic part.
>
> **Pseudo label of the AC Classifier:** As shown in Table A5, assigning pseudo labels in Eq. 6 was determined experimentally.
>
> The purpose of the AC classifier is to facilitate feature learning with discrepancies across domains. Therefore, when the gap between the abnormal scores of multiple heads exceeds a threshold $\tau$, it indicates different definitions of normal and abnormal between domains (e.g., abnormal in one domain and normal in another), and a conflict is considered to have occurred. When the score gap is small, it indicates consistent scores with no conflict, and the AC classifier generates a pseudo label accordingly.
>
> In Fig. 2, the yellow score graph shows a large gap between scores $S^a_{D_2}$ and $S^a_{D_M}$​, indicating an abnormal conflict and assigning $Y^{AC}=1$ in Eq. 6. Conversely, the green score graph shows consistent scores $S^a_{D_i}$ across all domains (all normal or all abnormal), indicating no conflict and assigning $Y^{AC}=0$. To determine conflicts between domains, we conducted ablation studies comparing different methods, such as the difference between scores of multiple heads, Std. with τ = 0.1, and a fixed value .
>
> We kindly refer reviewer to Section 4.2.5, "Role of the AC Classifier," and Section 4.3, "Discussions." If there are any mistakes in the figures, we would appreciate it if you could point them out for corrections.
>
> # [W2] Additional Baselines
> |    |           |       |       |       |        |       |       |        |
> |:---:|:---------:|:-----:|:-----:|:-----:|:------:|:-----:|:-----:|:------:|
> |    | **Models**    |       |       |       | **Target**|       |       |        |
> |    |           | UCFC  | XD    | LAD   | UBIF   | TAD   | ST    | AVG.   |
> | **E1** | WSAL      | 76.47 | 78.35 | 75.44 | 86.41  | 85.62 | 82.44 | 80.79  |
> |    | WSAL+Ours | 76.90 | 78.59 | 76.17 | 87.83  | 81.89 | 86.27 | 81.27  |
> |    |           |       |       |       |        |       |       |        |
> | **E2** | WSAL      | 76.59 | 73.75 | 76.71 | 79.86  | 77.68 | 55.54 | 73.36  |
> |    | WSAL+Ours | 76.67 | 73.69 | 65.54 | 85.85  | 70.1  | 72.41 | 74.04  |
> |    |           |       |       |       |        |       |       |        |
> | **E3** | WSAL      | 75.77 | 72.18 | 66.45 | 82.57  | 75.05 | 78.05 | 75.01  |
> |    | WSAL+Ours | 76.81 | 72.38 | 59.51 | 83.49  | 77.6  | 74.57 | 74.06  |
> |    |           |       |       |       |        |       |       |        |
> | **E4** | WSAL      | 80.99 | 81.6  | 76.43 | 87.87  | 88.78 | 87.47 | 83.86  |
> |    | WSAL+Ours | 79.21 | 81.17 | 78.19 | 90.98  | 86.09 | 90.61 | 84.38  |
> |    |           |       |       |       |        |       |       |        |
>
> We conducted additional experiments using the WSAL [29] model as a baseline to validate our proposed method on the MDVAD benchmark's four protocols. The above table presents the results, where all settings are consistent with the other experiments in the paper. The results show the addition of multi-head learning with NullAng-MIL and the AC Classifier to the WSAL model brings performance gains which effectively operate across different baselines. Notably, in the E3 and E4, which shows the target adaptation of the pre-trained general model, the method yielded competitive results.
>
> We focus more on highlighting the necessity of MDVAD and raising the issue of AC. Our paper introduces a novel task along with a benchmark, evaluation protocols, and a baseline and emphasizes analysis. Therefore, aspects like baseline architecture design or utilizing a powerful backbone were not the primary focus. However, future research will delve deeper into studying more sophisticated baselines for resolving AC within the MDVAD setting.
>
> # [W3] VAD with virtual datasets [i, ii]
> We appreciate the thoughtful comment. Following the reviewer's suggestion, we have added an analysis using the virtual dataset UBNormal [i] (please refer to the general response above).
>
> PFMP [ii] proposed a method to utilize virtual data anomalies to reduce scene discrepancy with real-world data, addressing the issue of data scarcity in VAD. While this method can mitigate data scarcity, it still faces the problem of anomaly conflicts when the criteria for abnormal events are inconsistent across multiple datasets.
>
> The conducted experimental analysis on utilizing virtual datasets [i] and a discussion on related work [ii] will be included in Section E of the supplementary material.
>
> [ii] "Generating anomalies for video anomaly detection with prompt-based feature mapping." CVPR, 2023.

---

> > ### Comment · Reviewer_926M · 2024-08-11
> >
> > Thanks for the authors' detailed response. After reading their response, some of my concerns are addressed. However, it is still not very clear for me that how the proposed Abnormal Conﬂict classifer addressed the abnormal conﬂict problem. I would like to see more detailed explaination from the authors, and I am open to hear from other reviewers.

---

> > > ### Author Response · Authors · 2024-08-12
> > >
> > > We are pleased that our rebuttal has addressed the reviewer's concerns.
> > >
> > > # [Q1] How the AC Classifier Addresses the AC Problem
> > >
> > > The AC classifier helps the model **learn conflict-aware features**. To provide a clearer understanding, we would like to explain this through a detailed step-by-step procedure.
> > >
> > > We have designed our framework with **Domain-Agnostic Layers** that learn multi-domain general features, followed by **Multiple Heads** that predict abnormalities for each domain. When the Domain-Agnostic layers learn the ability to perform AC classification, **their features capture knowledge about whether the input snippets are related to AC or not**. From the perspective of the Heads, these features are divided between AC and non-AC in the feature space, **allowing the head to apply different criteria (decision boundaries) when distinguishing between normal and abnormal**. For instance, when classifying abnormalities, non-AC scenarios can be addressed more easily, while AC scenarios require a more careful exploration.
> > >
> > > As a result, **1)** models trained with the AC Classifier handle AC more effectively in multi-domain learning (Table 5) and adapt better to unseen targets (Table 7). **2)** The AC Classifier learning was experimentally designed and validated (Table A3) and **3)** the AC score plot demonstrates that the model has effectively learned to predict AC (Fig. 3(a) and Fig. A5).
> > >
> > > In summary, the AC Classifier plays a crucial role in enhancing the model's ability to address AC problems by learning conflict-aware features, thereby allowing the Heads to aware conflicts when classifying normal and abnormal instances, leading to more effective learning. Thank you for the valuable feedback, and we hope this response alleviates the reviewer's concerns.

---

> > > > ### Comment · Reviewer_926M · 2024-08-13
> > > >
> > > > Thanks for the authors' detailed response.
> > > >
> > > > More specifically, I am confused that why AC labels are generated according to Eq. (6), i.e., why we should set the label to 1 only when the margin between max and min scores is larger than $\tau$. Could you please clarify this?
> > > >
> > > > By the way, the math notations are a little bit confusing in the manuscript. Please clarify them in your updated illustration.

---

> > > > > ### Author Response · Authors · 2024-08-13
> > > > >
> > > > > We gratefully appreciate for providing thoughtful feedback on our paper.
> > > > >
> > > > > # [Q2] AC Labels
> > > > >
> > > > > When generating pseudo labels for the AC Classifier, we utilize the score gap between multiple heads:
> > > > >
> > > > > **When Abnormal Conflict occurs between domains:** The abnormal scores produced by each domain's head will vary. For example, as shown in Fig 1, *'a pedestrian on the road'* is considered normal in the UCFC dataset but abnormal in the TAD dataset. In this case, the abnormal score from the UCFC head $s^a_{UCFC}$ would be low, while the abnormal score from the TAD head $s^a_{TAD}$ would be high. Therefore, the larger the score gap between domains, the more likely an abnormal conflict has occurred. This is represented by the following equation:
> > > > >
> > > > > $y_i^{AC}= 1$ where $[\max_{i} s^a_{D_d, i}-\min_{i} s^a_{D_d, i}-\tau]_+>0$
> > > > >
> > > > > **When normal and abnormal are consistent across domains:** The score difference between each domain's head will be minimal. In scenario where all domains consider the event normal, they will all have low abnormal scores, and in the opposite case, they will all have high abnormal scores. Thus, the difference between the maximum and minimum scores will be small and AC label will assign by following equation:
> > > > >
> > > > > $y_i^{AC}= 0$ where $[\max_{i} s^a_{D_d, i}-\min_{i} s^a_{D_d, i}-\tau]_+\le 0$
> > > > >
> > > > > We conducted various experiments, including difference, Std. ($\tau = 0.1$), and a fixed value ($y_i^{AC}=1$ where $0.3<s^a_{D_d,i}<0.7$) approaches when assigning labels (Please refer to Supplementary Material Section F.1 with Table A3). Experimentally, we determined that using the score difference, as in Eq. 6, is the most effective method for generating labels for the AC Classifier.
> > > > >
> > > > > # [Q3] Math notation
> > > > > We have organized the key notations below.
> > > > >
> > > > > * Abnormal video: $\\mathbf{V}^a\in \\left \\{ \\mathbf{v}^a_1, \cdots , \\mathbf{v}^a_i, \cdots , \\mathbf{v}^a_T \\right \\}$ where $i$ indicates snippet index.
> > > > > * Abnormal heads: $\\mathbf{w}^a_{D_d}\\in\mathbb{R}^{C/16\times1}$ where $d$ refers domain index
> > > > > * Abnormal score of $d$-th domain head: $\\mathbf{s}^a_{D_d}\\in\mathbb{R}^{T\times1}$ ($\\mathbf{s}^a_{D_d, i}$: $i$-th snippet abnormal score)
> > > > > * Abnormal score from NullAng: $\mathbf{s}^a_{D_d}= \mathbf{F} \cdot \mathbf{w}^a_{D_d} =\left \|| \mathbf{F} \right \||\left \|| \mathbf{w}^a_{D_d} \right \||\cos\boldsymbol{\theta}^a_{D_d}=\cos\boldsymbol{\theta}^a_{D_d}$ where final embedding feature is $\mathbf{F}$
> > > > > * Subset of Top-$K$ snippets' abnormal score: $\\Omega_k(\mathbf{s}^a_{D_{d}})$
> > > > > * AC score: $s^{AC}$ ($s^{AC}_i$: $i$-th snippet AC score)
> > > > >
> > > > > We hope our explanation has clarified any confusions, and we will also provide detailed descriptions of the mathematical notation in the Supplementary Material to ensure clarity.

---

> ### Comment · Reviewer_926M · 2024-08-13
>
> Thanks for the authors' detailed response.
>
> According to your illustration, in my understanding, the notation "$\max_{i} S^a_{D_d, i}$" should be "$\max_{d} S^a_{D_d, i}$" (also "$\min_{i} S^a_{D_d, i}$" should be "$\min_{d} S^a_{D_d, i}$"). This is because **the index $d$ denotes a domain and $i$ denotes a snippet**, and you argue that an abnormal conflict has more likely occurred as **the score gap between domains** is larger. I think there may be some typos in the formulas. Please check this.

---

> > ### Author Response · Authors · 2024-08-13
> >
> > We apologize for the confusion caused by typo in our notation. As the reviewer correctly pointed out, **the notation should be $d$ instead of $i$ in Eq. 6.** We will revise the equation in the manuscript accordingly.
> >
> > $y_i^{AC}= 1$ where $[\max_{d} s^a_{D_d, i}-\min_{d} s^a_{D_d, i}-\tau]_+> 0$
> >
> > $y_i^{AC}= 0$ where $[\max_{d} s^a_{D_d, i}-\min_{d} s^a_{D_d, i}-\tau]_+\le 0$
> >
> > Once again, we appreciate reviewer for pointing out this typo.

---

> > > ### Comment · Reviewer_926M · 2024-08-14
> > >
> > > Thanks for the authors' clarification. My concern about the design of abnormal conflict classifier has been addressed.
> > >
> > > In my understanding, this work proposes a new task, designs a new benchmark, and develop an interesting solution to address the task. Although this work has some flaws, I believe it makes a good contribution to the field, with its strengths outweighing its weaknesses. Therefore, I will increase my rating to borderline accept.
> > >
> > > By the way, I strongly encourage the authors to thoroughly check their manuscript and polish their writing, regardless of whether this paper is accepted.

---

### Official Review · Reviewer_beRv · 2024-07-08

**Soundness:** 3
**Presentation:** 3
**Contribution:** 2
**Rating:** 5
**Confidence:** 5

**Summary:**

In this paper, authors proposed a new task called Multiple Domain VAD (MDVAD), along with a benchmark and new evaluation protocols. Authors' goal is to construct a general VAD model by conducting multi-domain learning while recognizing abnormal conflicts and exploring representations of general normality and abnormality. Authors introduced a baseline for MDVAD and proposed a new framework with multi-head to mitigate abnormal conflicts and proposed Null-Multiple Instance Learning (Null-MIL) and NullAngular-MIL (NullAng-MIL) losses for multi-domain training. Additionally authors suggested an Abnormal Conflict (AC) Classifier to explore general features while being aware of abnormal conflicts. Authors analyzed the primary issues of MDVAD and proposed a baseline for this new task.

**Strengths:**

1. According to the analysis, authors believed that the abnormal conflict and the scene discrepancy are the two main issues and designed a framework with multi-head to deal with these problems.

2. Null-MIL and NullAng-MIL methods are designed for multi-domain learning, and an AC classifier is proposed for learning general features while abnormal conflicts exists.

3. Authors provided sufficient experiment results for this task and create a new baseline.

**Weaknesses:**

1. The proposed framework with multi-head for multi-domain seems not flexible enough during the domain changes, such as adding a new dataset with extra abnormal conflicts. And for the abnormal conflicts, will the proposed method performs better compared to make anomaly categories classifications for all anomaly events type of all domains?
2. In my opinion, traditional WS-VAD methods are designed to detect abnormal events in single domain without abnormal conflicts, and when abnormal conflicts exists, it will be better to use other paradigms such as temporal action localization or video grounding. And for the current WS-VAD datasets, the annotations are video-level, or even without category information, which is too weak for higher level anomaly detection. Training model with the current MDVAD paradigm is likely to not achieve good results.
3. Maybe using visual-language model with multimodal alignment can deal with the above issues? These models contain more knowledges for more event categories and higher generalization ability, which are likely to have the ability to individually detect conflicting anomalies. Compared to multi-head regression, is VL alignment a better approach for MDVAD task?

**Questions:**

My main questions are shown in the weakness.

---

> ### Author Rebuttal · Authors · 2024-08-07
>
> # [W1-1] when adding a new dataset
> Thank you for highlighting this important consideration. Our framework consists of domain-agnostic layers and domain-specific heads, with each head being the final layer of the entire model, $W_{D_d}\in \mathbb{R}^{T\times 1}$ where $T=128$, which is a very small part. If a new domain is introduced and additional models need to be trained, we can flexibly add a final branch. Creating a single general model is more cost-effective than developing individual in-domain models for each domain and retraining them every time a new domain appears. Moreover, such as in the E3 and E4 settings, pre-training the general model (E2) with multiple source domains without target domains, when new target data is added, can adapt to the new domain by tuning the source heads, without adding new heads. We will include this perspective in Section 4.3 discussion.
>
> # [W1-2] Abnormal classification for AC
> Even with comprehensive information on all abnormal categories across all domains, the lack of consistency in abnormality criteria between domains would still result in abnormal conflicts, posing challenges for abnormal classification through multiple domain learning. Additionally, in real-world scenario, addressing the issue through classification would lead to a closed-set model, limiting its ability to handle unexpected anomalies. We will discuss this approach in Section E of the supplementary material.
>
> # [W2] WS-VAD limitations and alternative paradigms
> We appreciate the reviewer's insightful comments. We would like to address each point raised:
> 1. **Traditional WS-VAD Methods in Single Domain**
> While traditional WS-VAD methods are designed for single-domain anomaly detection without conflicts, the MDVAD approach specifically addresses the limitations of these methods, which are heavily influenced by the criteria for abnormality defined by each dataset. Models that perform well in in-domain settings require each domain to be learned separately, necessitating sufficient training data for each single domain. This limits the applicability of anomaly detection methods to real-world settings. Therefore, research on a general VAD model capable of multiple domain learning is necessary to effectively mitigate the challenges posed by abnormal conflicts across different domains.
> 2. **Alternative Paradigms for AC (Temporal Action Localization or Video Grounding)**
> We agree that temporal action localization and video grounding are effective paradigms for certain types of video analysis. However, even with precise temporal annotations or detailed category information, defining anomalies precisely across multiple domains is challenging due to the varying criteria and sensitivity across datasets. This makes it difficult to provide a clear solution for conflicts.
> We hope this addresses your concerns. Thank you for your valuable feedback. Please let us know if any further considerations are needed.
>
> # [W3] Utilizing VL models
>
> We thank the reviewer for their thoughtful suggestion regarding the use of visual-language models with multimodal alignment. We appreciate the opportunity to discuss the potential advantages and considerations of such an approach in MDVAD task.
> * **Advantages of Visual-Language Models**: Recent studies have demonstrated the powerful capabilities of VLMs, achieving performance gains by leveraging VLM backbones and multimodal alignment for VAD tasks. Additionally, the integration of text information allows for meaningful interpretations of anomalies.
> * **AC detection with VL models**: While VLMs offer significant advantages, they also have limitations in capturing complex scenes in surveillance videos. According to [iii], existing VLMs may face challenges such as inaccuracies in color recognition, difficulty identifying intricate scenes, and struggles in capturing subtle movements. Furthermore, the availability of textual descriptions or annotations for VAD datasets is often limited, making high-level information like abnormal conflict detection an additional challenge.
> * **Future Directions**: We acknowledge the potential of VLMs and multimodal alignment as a promising direction for future research. Incorporating such models could enhance the ability to detect and classify a broader range of anomalies, especially in scenarios where textual annotations are available. For future work, we plan to explore this avenue to further improve the robustness and generalization capabilities of our MDVAD framework.
>
> [iii] Yuan, Tongtong, et al. "Towards Surveillance Video-and-Language Understanding: New Dataset Baselines and Challenges." Proceedings of the IEEE/CVF Conference on Computer Vision and Pattern Recognition. 2024.

---

### Official Review · Reviewer_SwSy · 2024-07-09

**Soundness:** 3
**Presentation:** 3
**Contribution:** 3
**Rating:** 5
**Confidence:** 4

**Summary:**

The manuscript addresses the limitations of existing Video Anomaly Detection (VAD) models that are confined to single-domain learning. The primary contribution of the paper is the introduction of a new task called Multi-Domain Learning for VAD (MDVAD), which aims to develop a general model capable of identifying abnormal events across multiple domains. The manuscript conducts experiments using the MDVAD benchmark and demonstrates the limitations of traditional multi-domain learning. It shows the effectiveness of the proposed baselines in handling abnormal conflicts and achieving robust performance across multiple domains.

**Strengths:**

1.The manuscript proposes a new task, Multiple Domain Video Anomaly Detection (MDVAD), which solves the problem that the existing model is limited to a single domain and provides a new idea for the development of domain-generalized models.
2.The MDVAD method proposes domain-specific multiple head mechanism and Null-Multiple Instance Learning Method (Null-MIL), which effectively solves the problem of anomaly conflict between different domains.
3. The MDVAD method constructs a new benchmark containing six representative VAD datasets, which fills the gap of the lack of unified evaluation standard in multi-domain learning tasks.
4. The MDVAD method designs four evaluation protocols (held-in, leave-one-out, low-shot domain adaptation, and full fine-tuning) to systematically evaluate the generalization ability of the model.

**Weaknesses:**

1. MDVAD introduces the domain-specific multi-head mechanism and the Null-MIL method, which increases the complexity and computational cost of the model, and may place higher demands on the computational resources in practical applications.
2. The multi-domain learning task itself is difficult to train, and with the proposed method further increasing the complexity of training, MDVAD may require longer training time and higher technical requirements.
3. Although the theoretical background and analysis are provided, the theoretical basis and derivation process of some of the methods of MDVAD are slightly weak and need to be further explored and verified in depth. Part of the theoretical analysis is based on specific assumptions, and these assumptions may not be fully valid in practical applications, affecting the applicability of the theoretical analysis.
4. Although new benchmarks and assessment protocols are proposed, MDVAD lacks comparative experiments with other state-of-the-art methods, making it difficult to objectively assess the relative advantages of the proposed methods.

**Questions:**

1. What is the training difficulty of MDVAD? The introduction of domain-specific multi-head mechanism and Null-MIL method greatly increases the complexity and computational cost of the model, can it meet the real-time requirements in practical applications?
2. Have the MDVAD and evaluation protocols been subjected to comparative experiments with other state-of-the-art methods in order to objectively assess the relative advantages of the proposed methods?

**Limitations:**

See Weaknesses.

---

> ### Author Rebuttal · Authors · 2024-08-07
>
> # [W1, W2, Q1] Complexity and computational cost
>
> In our proposed framework, only the final layer, $W_{D_d}\in \mathbb{R}^{T\times 1}$ where $T=128$, corresponds to the head and is added based on M number of datasets ($T \times M$). This constitutes a very small parameter and computational load compared to the total weights of the VAD model. We kindly refer reviewer to Section 3.2 on Complexity.
>
> # [W2, Q1] Training difficulty and inference time
>
> Regarding difficulty of training convergence, the pseudo-labels of the AC classifier are assigned through multiple heads (Eq. 6), which may lead to lower label reliability in initial training. However, this issue resolves as training progresses.
>
> Comparing a single head and multiple heads (when 6 datasets), the training times are 2.68 and 2.81 hours, and the inference times are 0.158 and 0.164 ms per snippet, respectively, indicating a negligible increase in complexity.
>
> # [W3] Theoretical analysis and applicability
> We would like to elaborate on how the following points support the theoretical flow of our work:
> * **[Section 2.2 and Section D]** Our motivation and the necessity for the novel task MDVAD are derived from the analysis of the VAD benchmark and the cross-evaluation of single domain models (Section 2.2, Tables 1 and 2). We have identified two key issues: Abnormal Conflict and Scene Discrepancy. We validated our assumption regarding AC by computing $AC_{i,j}$ using Eq. 8, which measures the average of the relative False Positive Rate (FPR) and relative False Negative Rate (FNR) to quantify abnormal conflict between domains (Supplementary Material, Section D, Table A2). For the domain discrepancy assumption, we analyzed the Earth Mover’s Distance (EMD) (Table 3).
> * **[Section 4]** We defined protocols for four scenarios (E1~E4) considering practical applicability and conducted quantitative and qualitative experimental analyses to support our assumptions about AC and domain discrepancy:
> E1: When all source and target domains are accessible, performing similar to multi-task learning, handling multiple domains with a single model.
> E2: When the source and target domains are different, and the target is not accessible, pre-training on large data without target knowledge simulates the scenario where training data for the practical application domain is unavailable.
> E3 : When a few samples from the target domain are provided, adapting the pre-trained general model.
> E4: When the pre-trained general model is fine-tuned for the specific needs of the target domain.
>
> # [W4, Q2] Comparison experiments
> |    |        |          |       |       |       |        |       |       |        |   |   |
> |:---:|:------:|:--------:|:-----:|:-----:|:-----:|:------:|:-----:|:-----:|:------:|:---:|:---:|
> |    | **Models** | **Pub.**     |       |       |       | **Target** |       |       |        |   |   |
> |    |        |          | UCFC  | XD    | LAD   | UBIF   | TAD   | ST    | AVG.    |   |   |
> | **E1** | MMIL   | CVPR 18' | 77.93 | 81.34 | 85.18 | 85.44  | 87.78 | 84.39 | 83.68  |   |   |
> |    | ARNet  | ICME 20' | 79.26 | 80.38 | 85.27 | 84.18  | 89.57 | 86.65 | 84.22  |   |   |
> |    | WSAL   | TIP 21'  | 76.47 | 78.35 | 75.44 | 86.41  | 85.62 | 82.44 | 80.79  |   |   |
> |    | COMO   | CVPR 23' | 80.41 | 82.75 | 86.24 | 85.82  | 90.13 | 89.76 | 85.85  |   |   |
> |    | **Ours**   |     -     | 77.21 | 82.09 | 83.88 | 91.9   | 91.36 | 91.12 | **86.26**  |   |   |
> |    |        |          |       |       |       |        |       |       |        |   |   |
> | **E2** | MMIL   | CVPR 18' | 76.68 | 74.92 | 67.39 | 82.4   | 67.61 | 61.86 | 71.81  |   |   |
> |    | ARNet  | ICME 20' | 77.05 | 75.02 | 78.98 | 80.84  | 75.09 | 55.34 | 73.72  |   |   |
> |    | WSAL   | TIP 21'  | 76.59 | 73.75 | 76.71 | 79.86  | 77.68 | 55.54 | 73.36  |   |   |
> |    | COMO   | CVPR 23' | 77.07 | 76.64 | 77.43 | 76.74  | 78.67 | 57.63 | 74.92  |   |   |
> |    | **Ours**   |     -     | 78.55 | 77.68 | 77.36 | 82.53  | 79.21 | 60.41 | **75.96**  |   |   |
> |    |        |          |       |       |       |        |       |       |        |   |   |
> | **E3** | MMIL   | CVPR 18' | 77.28 | 72.75 | 80.7  | 85.29  | 81.72 | 61.33 | 76.51  |   |   |
> |    | ARNet  | ICME 20' | 75.48 | 72.18 | 79.9  | 81.7   | 79.43 | 68.24 | 76.16  |   |   |
> |    | WSAL   | TIP 21'  | 75.77 | 72.18 | 66.45 | 82.57  | 75.05 | 78.05 | 75.01  |   |   |
> |    | COMO   | CVPR 23' | 70.02 | 72.89 | 80.59 | 83.05  | 74.73 | 73.53 | 75.80  |   |   |
> |    | **Ours**   |    -      | 78.99 | 75.8  | 77.82 | 85.75  | 84.06 | 76.23 | **79.78**  |   |   |
> |    |        |          |       |       |       |        |       |       |        |   |   |
> | **E4** | MMIL   | CVPR 18' | 80.26 | 82.51 | 86.54 | 89.88  | 90.32 | 89.34 | 86.48  |   |   |
> |    | ARNet  | ICME 20' | 80.88 | 82.57 | 86.72 | 89.92  | 90.69 | 91.7  | 87.08  |   |   |
> |    | WSAL   | TIP 21'  | 80.99 | 81.6  | 76.43 | 87.87  | 88.78 | 87.47 | 83.86  |   |   |
> |    | COMO   | CVPR 23' | 80.61 | 84.25 | 86.88 | 91.51  | 91.74 | 91.23 | **87.70**  |   |   |
> |    | **Ours**   |     -     | 78.62 | 82.71 | 84.41 | 94.42  | 92.5  | 91.17 | 87.31  |   |   |
> |    |        |          |       |       |       |        |       |       |        |   |   |
>
> Instead of exploring complex architecture designs for single-domain VAD models, this paper focuses on the analysis of multiple domain learning within the context of AC issues. In response to the reviewer's concerns, we compared MDVAD with four representative WVAD models (Section E): MMIL [42], ARNet [45], WSAL [29], and CoMo [7]. In the Table, proposed simple baseline demonstrates competitive performance. Notably, the average results show superior results in the E2 and E3, indicating better generalization and adaptation to unseen target domains. Various single-domain VAD models or backbones can be incorporated into the MDVAD task, showing a direction for future generalization work (please refer to the W2 response of 926M).

---

> > ### Comment · Reviewer_SwSy · 2024-08-13
> >
> > I consider the methodology of this work to be another innovative approach to anomaly detection that is different from previous methods. Although there are some flaws in the work, it is a good starting point. I wish there were more types of approaches to anomaly detection. I will keep my marks. Good luck!

---

### Official Review · Reviewer_2cHp · 2024-07-12

**Soundness:** 2
**Presentation:** 2
**Contribution:** 2
**Rating:** 4
**Confidence:** 4

**Summary:**

This paper proposes a new task called MDVAD, the goal of which is to effectively learn from multiple domains with different data distributions and definitions of abnormality without confusion, resulting in a general VAD model. To achieve this, the authors expand the traditional single-head framework to multiple-head framework for learning different knowledge and design an AC classifier to handle abnormal conflicts. The experimental results prove the effectiveness of the proposed method.

**Strengths:**

1. This paper focuses on the problem of learning a generalizable VAD model, which is an important task.
2. The experiments conducted by the author are relatively compr

**Weaknesses:**

1. This paper proposes a new task called MDVAD to achieve generalizable VAD by resolving conflicts in anomaly definitions. However, for any VAD application, the definition of normal or abnormal events should be explicitly determined according to the scenario requirements, rather than simply combining multiple datasets and resolving the abnormal conflicts. I find it difficult to understand under what practical scenario a VAD model trained using multiple datasets with abnormal conflicts is needed.
2. The writing of this paper is not clear enough, where some necessary training and inference details are missed. For example, the normal head training mentioned in NullAng-MIL is confusing.
3. This paper lacks a detailed description of the experimental setup. For example, if an anomalous event is determined to be a conflict, how should the model handle such an event?

**Questions:**

1. (referring to weakness 1) In practical applications, what kind of scenarios conform to the task settings of MDVAD proposed by the authors?
2. (referring to weakness 3) During test phase, is it necessary to know which dataset the sample comes from? If a certain anomalous event is found to have conflicts in different datasets, how should it be handled? Do the multi-dataset evaluation and the test procedures for other methods use the same test data?

**Limitations:**

I do not recognize obvious potential negative societal impact of this work.

---

> ### Author Rebuttal · Authors · 2024-08-07
>
> # [W1, Q1] Practical scenarios of MDVAD
>
> **MDVAD’s practical relevance**
> In real-world scenarios, performance degradation due to domain shift is a persistent issue for deep learning models. Consequently, various tasks have seen the introduction of domain adaptation and generalization methods. As reported in cross-domain evaluation performance studies [6, 7, 24], VAD models experience more than a 20~60% drop in AUC, largely due to the criteria for abnormality defined by each dataset. Models that perform well only within a single domain are significantly limited in their practical applicability.
> Additionally, collecting data for every possible scenario in real-world applications is challenging, and abnormal events occur rarely, resulting in data scarcity. To address these challenges, our approach leverages multiple VAD datasets and aims to learn a general model that effectively mitigates the ambiguity in defining abnormalities across datasets.
>
> **Utilizing a generalizable VAD model**
> A general VAD model trained on multiple domains offers two key benefits:
>
> * First, when data from the target domain in the real world is available (E1), a general model trained on multiple source datasets, including the target domain, is able to explore robust and general features, similar to the effects of multi-task learning. Additionally, a single generalized model eliminates the need for multiple specific models for different domains.
> * Second, when target domain training data is not available (E2), proper pre-training on multiple domains allows the general VAD model to embody generalized representations, leading it to better performance in unseen target domains. Furthermore, when a real-world target dataset is provided (E3:, E4), the pre-trained general VAD model can adapt well to the new domain, which is highly beneficial for practical VAD applications.
> We are grateful for the reviewer’s insightful comment and will incorporate this discussion on this aspect in Section 4.3 of our paper.
>
> # [W2] Training/Inference details of normal head in NullAng-MIL
>
> NullAng-MIL uses an angular margin to learn normal and abnormal features effectively. It adjusts the feature vector distances in the angular space both inter- and intra-class within each domain.
>
> Regarding the training of the normal head, when input training data is from domain $j$, the normal score$s^n $ is output from $j^{th}$ Normal head ${W}^n_{D_j}$ and  abnormal score $s^a $ is output from ${W}^a_{D_j}$.
>
> ${s}^n_{D_j}=F \cdot {W}^n_{D_j}=\left\| F \right\|\left\| {W}^n_{D_j} \right\|cos{\Theta }^n_{D_j}$
>
> In the equation, $F$ is the final embedding feature. By normalizing both the feature and the head, score $s$ is the angle in the angular space between the two vectors. For simplicity, the snippet index $i$ is omitted. When $F$ is the feature of a normal snippet, the angle with the normal head${W}^n_{D_j}$ should be narrower than the angle with the abnormal head. Conversely, when $F$  is from an abnormal snippet, the opposite should hold true.
>
> $\begin{cases}cos{\Theta }^a_{D_j} + m > cos{\Theta }^n_{D_j} & \textrm{when abnormal snippet}\\\cos{\Theta }^a_{D_j} < cos{\Theta }^n_{D_j}+m & \textrm{when normal snippet}\end{cases} $
>
> Therefore, by adding the angular margin $m$, the feature learning process is designed to satisfy the above conditions. When training the normal head, other normal heads $\\{{W}^n_{D_d}\\}$ where $d \neq j$ do not affect the gradient, which is calculated as $\frac{ \partial  s^n}{ \partial {W}^n_{D_j}}$.
>
> # [W3, Q2] Missing detail description
>
> **Experimental setup**
> The four protocols to verify the MDVAD task are as follows:
>
> * **E1 Held-in**: Integrating various domain’s anomalies into one framework, performing similarly to multi-task learning that handles multiple tasks with a single model. All source and target domains are accessed. A single model trained on $M$ (M=6) domains is evaluated on $M$ target domains.
> * **E2 Leave-one-out**: A methodology of pre-training on large data without target knowledge. The source and target are different, and the target is not accessed. Training with $M-1$ source domains in a held-out setting and applying to an unseen target dataset.
> * **E3 Low-shot adaptation**: Evaluates the adaptation ability of the general pre-trained model (E2). When few samples (10% of the training set) from the target domain are given, low-shot learning is performed, and the model is evaluated on the target domain.
> * **E4 Full fine-tuning**: Evaluates the performance of the general pre-trained model (E2) after full finetuning on the target domain’s training set.
>
> **Handling abnormal conflict**
> When performing multiple domain learning, inconsistent labels across domains can lead to abnormal conflicts, causing confusion in the model's training and making it difficult to develop a robust general model. Our proposed framework consists of domain-agnostic layers and multiple heads for  different domains. While the heads are divided to prevent conflicts, the agnostic part extracts features from all datasets using a single branch. Therefore, to explore general features while being aware of abnormal conflicts, an auxiliary branch, the AC classifier, predicts conflicts by leveraging the variance in abnormal scores across multiple heads.
>
> **Testing phase** During the testing phase, target domain information is unnecessary in the E2~E4 settings because the source domain of the pre-trained general model is different from the target domain, it does not have a head for the target domain. As in Eq. 5, under the E1 setting, since we have information about the target domain, the target branch score is used as the final score. As in Eq. 5, we determine the final score with both normal and abnormal score to reflect conflicts by taking the maximum normal and abnormal score from multiple heads. All empirical studies of the MDVAD and the comparison models were evaluated and reported on the same source and target domains.

---

> > ### Comment · Reviewer_2cHp · 2024-08-13
> >
> > I greatly appreciate the authors' responses and the additional experiments, which largely addressed my concerns. My comments about their responses are listed as follows.
> >
> > W1: The authors mention its application scenario as: "During the testing phase, target domain information is unnecessary in the E2~E4 settings because the source domain of the pre-trained general model is different from the target domain." However, I still remain skeptical about its practical significance.
> >
> > W2: The authors have re-elaborated on the training methodology, and the overall process is relatively clear.
> >
> > Based on the above considerations, I still have doubts regarding the practical relevance of the task proposed by the author. As such, I will maintain my current score for the time being and am open to hearing the opinions of other reviewers. If my doubts are resolved through further discussion, I will not refuse to increase my score accordingly.

---

> > > ### Author Response · Authors · 2024-08-13
> > >
> > > We are pleased that our rebuttal has largely addressed the reviewer's concerns. We sincerely appreciate reviewer's thoughtful comment.
> > >
> > > # [W1] Practical Applications
> > >
> > > **Problem Definition of Domain Generalization**
> > >
> > > The goal of Domain Generalization (DG) is to learn a robust and generalizable predictive function from the $M$ training domains to achieve minimal prediction error on an unseen test domain that cannot be accessed during training. The difference between **Domain Adaptation (DA) and DG is that DA has access to the target domain data while DG cannot see them during training. This makes DG more challenging than DA but more realistic and favorable in practical applications [R1].** Various approaches, such as zero-shot learning (E2), adaptation learning (E3, E4), meta-learning, lifelong learning, and transfer learning, are employed to address the emergence of unknown target domains in real-world scenarios.
> > >
> > > Foremost, we would like to clarify that the focus is on the fact that the **“Even without Target Domain information, the model performs comparably to an In-Domain (Single) Model”**, rather than on the idea that “Target Domain information is unnecessary.” In real-world scenarios that an unknown target domain emerges, the pre-trained general model learned from multiple source domains can operate using Eq. 5 without Target Domain information. Moreover, if samples from the target domain are provided, the model can adapt by tuning the multiple heads.
> > >
> > > As mentioned previously, **the ability to handle multiple domains with a single model (E1) is also a practical scenario for generalization**. Because requiring a separate model and training for each domain is neither practical nor robust.
> > >
> > > Similarly, issues such as Domain Shift, Overfitting, Label Conflict, and Efficiency are all challenges that need to be addressed for practical applications.
> > >
> > > We acknowledge that our paper represents the first step toward practical VAD applications, and it opens up the possibility for various future works that could lead to valuable applications. We hope that the reviewer's concerns have been resolved, and we will include a discussion on this practical perspective in the manuscript.
> > >
> > > [R1] Wang, Jindong, et al. "Generalizing to unseen domains: A survey on domain generalization." IEEE transactions on knowledge and data engineering 35.8 (2022): 8052-8072.

---

### Author Rebuttal · Authors · 2024-08-07

We thank the reviewers for their thoughtful feedback. We tried to address all the questions with references to weaknesses (**W**) and questions (**Q**).

We are glad the reviewers found that
* Addressing an Important and Generalizable Problem
* Novelty of Method and Effectively Solving Identified Problems
* Thorough Experiments and a Systematic Protocols for Model Generalization

# Summary of rebuttal
In response to the reviewers’ feedback, we will improve the paper as follows:
- We have added additional virtual UBNormal datasets for multiple-domain learning and open-set VAD experiments with discussions (reviewer 926M).
- We have expanded experiments on additional baseline and state-of-the-art comparison methods with detailed discussions (reviewer SwSy and 926M).
- We have provided clear explanations and detailed descriptions regarding the theoretical analysis, practical applicability, experimental settings, and the role of the AC classifier (reviewers SwSy, 2cHp, and 926M).
- We have included further discussion about other alternative paradigms and future work with vision-language models (reviewer beRv).

# Results added using virtual data
In response to Reviewer 926M's comment regarding the discussion on utilizing virtual datasets, we have included the UBNormal dataset [i] alongside the MDVAD to conduct multi-domain learning and validate the model's generality in settings with significant scene discrepancy.

1. **Multi-domain learning with virtual domain dataset:**

|                     |       |       |        |       |       |       |       |
|---------------------|-------|-------|--------|-------|-------|-------|-------|
| **Baseline**           |       |       | **Target** |       |       |       |       |
|                     | UCFC  | XD    | LAD    | UBIF  | TAD   | ST    | AVG.  |
| **Source: MDVAD**       |       |       |        |       |       |       |       |
| MIL                 | 80.05 | 83.77 | 86.01  | 85.76 | 88.92 | 88.82 | 85.56 |
| MIL+AC              | 80.11 | 83.91 | 85.15  | 87.72 | 90.05 | 87.98 | 85.82 |
| NullMIL             | 79.01 | 81.96 | 85.08  | 93.06 | 90.57 | 91.04 | 86.79 |
| NullMIL+AC          | 79.15 | 82.96 | 85.82  | 92.41 | 91.16 | 89.67 | 86.86 |
| NullAngMIL          | 76.32 | 82.74 | 82.32  | 92.30 | 91.82 | 91.26 | 86.13 |
| NullAngMIL+AC       | 77.21 | 82.09 | 83.88  | 91.90 | 91.36 | 91.12 | 86.26 |
|                     |       |       |        |       |       |       |       |
| **Source: MDVAD + UBN** |       |       |        |       |       |       |       |
| MIL                 | 78.59 | 81.79 | 85.06  | 86.6  | 88.83 | 87.99 | 84.81 |
| MIL+AC              | 78.14 | 81.78 | 84.92  | 85.08 | 90.77 | 89.12 | 84.96 |
| NullMIL             | 78.74 | 83.34 | 85.93  | 91.28 | 90.05 | 88.69 | 86.34 |
| NullMIL+AC          | 78.76 | 83.19 | 86.01  | 92.63 | 90.73 | 90.54 | 86.98 |
| NullAngMIL          | 77.09 | 81.01 | 83.96  | 92.88 | 91.57 | 90.04 | 86.09 |
| NullAngMIL+AC       | 77.66 | 82.09 | 83.01  | 92.55 | 91.33 | 91.07 | 86.29 |
|                     |       |       |        |       |       |       |       |

The UBNormal (UBN) dataset is a VAD benchmark proposed for open-set scenarios to handle unexpected abnormal events. Both normal and abnormal events are available during training, but the anomalies that occur during inference belong to a distinct set of anomaly types (categories). To alleviate the difficulty of collecting abnormal event data in the real world, UBN consists of synthetic videos, unlike other VAD datasets. There are substantial abnormal conflicts and differences in the visual settings of scenes compared to other domains.

The table shows the performance in each target domain when trained with MDVAD and UBNormal under the E1: held-in setting. In the single-head MIL baseline, there is a performance drop when trained with MDVAD+UBN, indicating difficulty in handling Abnormal Conflict (AC). However, the model trained with multiple heads and the AC classifier shows improved results. By leveraging the virtual dataset, we can overcome data limitations and create a general model capable of handling diverse and complex scenes.

2. **Openset VAD results**

|                            |                |
|----------------------------|----------------|
| **Open-set VAD**               |                |
| **Single source:**       UBN         |  / **Target:** UBN |
| MIL                        | 75.13          |
|                            |                |
| **Multi source:** MDVAD + UBN  | / **Target:** UBN  |
| MIL                        | 67.95          |
| MIL+AC                     | 70.56          |
| NullMIL                    | 72.14          |
| NullMIL+AC                 | 70.94          |
| NullAngMIL                 | 74.42          |
| NullAngMIL+AC              | 74.54          |
|                            |                |

We conducted experiments in an open-set scenario using UBN, where the abnormal categories in the train set and test set do not overlap. As shown in the table, despite domain discrepancies and AC, the model effectively handles multiple domain learning, demonstrating that general feature learning can adequately address unseen abnormal categories.

[i] "Ubnormal: New benchmark for supervised open-set video anomaly detection." CVPR. 2022.

---

### Decision · Program_Chairs · 2024-09-25

**Decision:**

Accept (poster)

**Comment:**

This paper received 4 reviews from experts in the field. The paper received the following reviews:  3 Borderline Accepts and 1 Borderline Reject.

The main concerns presented by the reviewers involved the clarity of the presentation. The author rebuttal addressed the reviewers’ concerns. The AC agrees that the merits of this paper outweigh any clarity concerns and the decision for this paper is to accept.  Please consider the reviewer comments in your preparation of the camera ready version of the paper.